# LOOK BEFORE YOU LEAP: ESTIMATING LLM BENCHMARK SCORES FROM DESCRIPTIONS

## ABSTRACT

Progress in large language models is constrained by an evaluation bottleneck: build a benchmark, run models, then iterate. We ask a question: *can we forecast outcomes before running any experiments to inform earlier study design*? For example, a team building an AI assistant for a certain task can estimate whether expected performance is around 50 or closer to 80, evidence that supports whether to proceed to a pilot study, how to scope it, and how to allocate resources. We study text-only performance forecasting, where a model predicts a score from a redacted task description and intended configuration, with no access to dataset instances. To support systematic study, we curate PRECOG, a corpus of redacted description–performance pairs spanning diverse tasks, domains, and metrics. We scrape task and configuration descriptions from arXiv, yielding 2,290 instances covering 1,519 papers, and construct a leakage-free test split using papers published after the evaluated models' knowledge cutoff. Experiments show the task is challenging but feasible: reasoning models achieve moderate prediction performance with well-calibrated uncertainty, reaching mean absolute error as low as 9.9 at high-confidence thresholds. We further test a zero-leakage setting—forecasting on newly released datasets or experiments before their papers are indexed—where GPT-5 with built-in web search still attains non-trivial prediction accuracy. Overall, our corpus and analyses offer an initial step toward open-ended anticipatory evaluation, supporting difficulty estimation and smarter experiment prioritization.

## 1 INTRODUCTION

Modern LLM development is evaluation-centric: define a benchmark, evaluate models and configurations, iterate (Brown et al., 2020; Achiam et al., 2023; Touvron et al., 2023; Team et al., 2023). As benchmarks proliferate, exhaustive curation and full experiments demand substantial compute, researcher time, and funding (Mazumder et al., 2023). We therefore seek an *ex ante* performance forecaster to help designers decide what to build and where to measure. For example, consider a team proposing a binary "solver-choice" task on an ordinary differential equations benchmark (Figure 1; Gaonkar et al., 2025). Before annotating a single example, they can ask a concrete what-if: will GPT-4.1 (Achiam et al., 2023) without CoT (Wei et al., 2022) land nearer 30 or 60 Accuracy points? A forecast like this has the potential to quickly inform scope and spend—how large a pilot to run, which baselines to include, and whether the task merits full-scale annotation.

Two adjacent lines of work on ML model performance prediction address related goals: (i) *missing-entry prediction*, which infers unobserved cells from observed ones (Xia et al., 2020; Ye et al., 2021; 2023; Zhao et al., 2025); and (ii) *efficient evaluation*, which extrapolates full-benchmark performance from a small set of pilot runs (Maia Polo et al., 2024; Pacchiardi et al., 2024; Anugraha et al., 2025). Both assume seed measurements or closely related tasks and typically rely on engineered numeric features. In contrast, we target a *zero-example, open-ended* setting in which retrieval is driven by natural-language task descriptions, supporting researchers who are creating new datasets or designing new experiments. Prior work has also explored the capability of LLMs to forecast the outcomes of future geopolitical events in online prediction markets (Karger et al., 2024; Halawi et al., 2024), however to the best of our knowledge, this is the first work to explore the capability of LLMs to predict their own performance on a new task based only on a text description.

**Estimate GPT-4.1's performance based on the following experimental description** 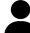

**Task**
Binary classification of solver type for ODEs: given an ODE expression, the model must decide whether to use an explicit or implicit numerical integrator. For this subset, the scientifically correct choice is always an explicit method.

**Data collection**
Synthetic diagnostic set constructed to probe symbolic simplification ability. Problems are designed to look numerically "stiff" due to large coefficients or misleading forms but become non-stiff after simplification.

**Evaluation**
Accuracy = proportion of problems for which the model selects an explicit solver.

**Difficulty characteristics**
Requires nontrivial symbolic manipulation rather than pattern matching. Superficially large coefficients that can mislead solver selection. Binary decision (explicit vs. implicit) with correctness contingent on successful simplification.

**Subset**
Trigonometric subset: instances simplify via trigonometric identities (e.g., angle-shift/sum relations) to reveal non-stiff dynamics, making an explicit method appropriate.

**Prompting strategy**
Base/unguided prompt: minimal instructions that ask the model to classify the ODE and choose a solver, without step-by-step reasoning guidance, algebraic-simplification hints, or in-context examples.

**Regression through reasoning with literature search** 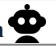

**Rationale**
- Symbolic-manipulation diagnostics reveal large performance drops under small perturbations.
- Zero-shot prompt tends to depress math performance: GPT-4's zero-shot accuracy on MATH is ~49%.
- Subset requires nontrivial trigonometric simplification; prior work (TRIGO) reporting struggling with trig-expression.
- Data are intentionally "stiff-looking". LLMs' math performance is known to degrade under superficial/wording variations.

**Point Estimate: 32** ⟷ **Target: 21.7** (Absolute Error: 10.3)

Figure 1: PRECOG pairs a redacted task/dataset description with its target metric. Given the inputs (a redacted description of dataset and an experimental configuration (Gaonkar et al., 2025)), the goal is to predict the target metric. The regressor conditions on performance-relevant details and may retrieve corroborating evidence from the literature.

We formulate text-only performance forecasting as regression: given a natural-language specification of the dataset and evaluation setup, including task, data provenance, prompting/decoding choices, and metric/protocol, we predict a target metric normalized to a 0–100 scale. As in Figure 1, the input is a description of the proposed task and configuration; the output is a predicted metric value (e.g., 32). Unlike feature-engineered baselines, we operate directly on free-form, literature-style text and optionally use retrieval to consult related prior work, rather than interpolation from seed results.

To enable systematic study, we curate PRECOG, an arXiv-mined corpus of experimental records (Park et al., 2025) that, for each entry, links the benchmark's source paper (introducing the dataset) to the paper reporting results on it. From these papers we extract task and configuration descriptions, yielding 2,290 instances spanning 1,519 papers, and we construct a leakage-free test split by restricting to papers published after the evaluated models' knowledge cutoff (2025 January-September). From these sources, we synthesize redacted descriptions that preserve task and setup details while masking dataset names and numeric results. To further probe residual leakage risks (Tian et al., 2024), we also evaluate a streaming regime that ingests arXiv releases and makes predictions before they are indexed by search engines.

We then evaluate zero-shot LLM regressors, optionally augmented with literature search. Given a redacted description, the model may iteratively issue evidence-seeking queries (ReAct-style (Yao et al., 2023)) and outputs a point estimate alongside a self-reported confidence (Yoon et al., 2025). We assess calibration by sweeping a confidence threshold and reporting the coverage–MAE trade-off, and we analyze search tool-use behavior.

On PRECOG, a proprietary reasoning model attains an MAE of $14.6$ across seven metric groups, and $9.9$ on a high-confidence, accuracy-filtered subset, with reasonably calibrated uncertainty. We perform both reasoning and subset ablations to identify which factors drive performance prediction and to characterize its behavior. In the streaming setting, GPT-5 (OpenAI, 2025) with built-in web search delivers competitive, though imperfect, performance on newly released datasets, while maintaining good calibration.

Altogether, our study shows that LLM-based regressors that predict model performance from text descriptions can inform benchmark design, prioritize configurations, determine dataset sizes, and plan pilot experiments, while reducing compute and turnaround time for exploratory analysis. We release PRECOG, a redacted-description benchmark grounded in the scientific literature, together with prompts, protocols, and code. We hope these resources and our evaluation methodology will support future advances in predictive evaluation and model selection.

## 2 RELATED WORK

**Performance Prediction** Prior works primarily fall into two categories. (1) *Missing-entry prediction*: infer unobserved cells from observed ones using matrix-completion or response-surface models (Achille et al., 2019; Xia et al., 2020; Chen & Ritter, 2021; Ye et al., 2021; 2023; Rabinovich et al., 2023; Shi et al., 2025; Zhao et al., 2025); these assume seed results, stable task identities, and overlap for interpolation. (2) *Efficient evaluation*: run a small subset of trials and extrapolate full-benchmark performance (Maia Polo et al., 2024; Pacchiardi et al., 2024; Anugraha et al., 2025); these still require executing target (or closely related) runs. Finally, we note contemporaneous work by Wen et al. (2025), which aims to predict which of two novel research ideas is likely to perform better on

| Work | Task Design | | | | | |
| | Text. | Feat. | Runs | Idea | Task | Reg. |
|---|---|---|---|---|---|---|
| Xia et al. (2020) | ✗ | ✓ | ✗ | ✗ | ✗ | ✓ |
| Ye et al. (2021) | ✗ | ✓ | ✗ | ✗ | ✗ | ✓ |
| Ye et al. (2023) | ✗ | ✓ | ✗ | ✗ | ✗ | ✓ |
| Maia Polo et al. (2024) | ✗ | ✗ | ✓ | ✗ | ✗ | ✓ |
| Pacchiardi et al. (2024) | ✗ | ✓ | ✓ | ✗ | ✗ | ✓ |
| Anugraha et al. (2025) | ✗ | ✓ | ✗ | ✗ | ✗ | ✓ |
| Wen et al. (2025) | ✓ | ✗ | ✗ | ✓ | ✗ | ✗ |
| PRECOG (ours) | ✓ | ✗ | ✗ | ✗ | ✓ | ✓ |

Table 1: Comparison of performance-prediction schemas. **Text.**: task descriptions; **Feat.**: numeric features; **Runs**: running the model on a subset of the benchmark; **Idea**: predicts unseen method's performance; **Task**: predicts performance on an unseen task; **Reg.**: (Regression) predicts an metric value.

a set of *existing benchmarks*. In contrast, our work demonstrates that LLMs can predict the absolute performance of a fixed model on a *novel benchmark*. See Table 1 for a detailed comparison.

**LLMs for Regression** LLMs have been used as regressors in structured settings (AutoML/black-box optimization) (Song et al., 2024; Vacareanu et al., 2024; Nguyen et al., 2024; Requeima et al., 2024; Tang et al., 2024), often relying on key–value hyperparameter strings and access to seed trials or finetuning on the target task. In contrast, we study a zero-run, description-only regression problem where the sole input is a free-form specification of a novel dataset and evaluation protocol. Our task involves forecasting, specifically, predicting the outcomes of ideas before conducting real experiments. (Zou et al., 2022; Schoenegger & Park, 2023; Karger et al., 2024; Abolghasemi et al., 2025; Moussa et al., 2025). Inspired by these works, our work demonstrates the potential of reasoning LLMs in forecasting the outcomes of evaluation results on unseen datasets.

**Extracting Structured Records from Litertaure** Prior work extracts structured result tuples (e.g., task, dataset, metric) from scientific articles (Singh et al., 2019; Hou et al., 2019; Kardas et al., 2020; Yang et al., 2022; Bai et al., 2023; Şahinuç et al., 2024; Park et al., 2025). PRECOG builds on these tuples: for each experimental record, we synthesize a richer natural-language description that augments the tuple with additional context.

## 3 TASK

**Formulation** We study zero-example, text-only performance forecasting as a regression problem. Given a self-contained, redacted description $x_i$ of a dataset and evaluation configuration, the model performs reasoning and outputs a numeric estimate $\hat{y}_i \in [0, 100]$. We treat $x_i$ as a structured yet free-form text description that captures contextual nuances of the task, dataset, and experimental configuration without requiring access to instances. To compare heterogeneous metrics (e.g., Accuracy, F1, BLEU), we convert each paper-reported score $y_i^{\text{paper}}$ to a common 0–100 scale via a simple normalization ($N : \mathbb{R} \to [0, 100]$); $y_i = N(y_i^{\text{paper}})$. Concretely, values already on a 0–100 scale remain unchanged, while fractional scores in ($[0, 1]$) are multiplied by 100 (e.g., Accuracy ($0.35 \mapsto 35.0$); F1 ($83.2 \mapsto 83.2$).

Given a redacted description $x_i$, an LLM regressor $f_\theta$ predicts the normalized target as $\hat{y}_i = f_\theta(x_i, Z_i)$, where $Z_i$ is an optional set of retrieved evidence. We obtain $Z_i$ by querying a time-bounded literature corpus $\mathcal{C}$ with a top-$k$ retriever $R_k$ on a query formed from $x_i$, i.e., $Z_i = R_k(q(x_i), \mathcal{C})$. Setting $k = 0$ yields the *description-only* variant $f_\theta(x_i, \varnothing)$. At inference, $f_\theta$ may iterate in a ReAct-style loop—reason (Yao et al., 2023), issue $q(\cdot)$, retrieve $Z_i$, refine—and then output $\hat{y}_i$. For example, for the open-domain QA description above, $f_\theta$ can retrieve prior exact-match (EM) ranges for similar setups and use them to refine $\hat{y}_i$.

**Evaluation** Following prior work (Srinivasan et al., 2021; Ahuja et al., 2022; Jawahar et al., 2024), we evaluate $f_\theta$ with two standard criteria: mean absolute error (MAE) and the Pearson correlation r between $y_i$ and $\hat{y}_i$. MAE (lower is better) measures absolute accuracy across heterogeneous metrics, while r (higher is better) assesses whether predictions preserve the relative ordering of tasks/configurations. Used together, they capture both value accuracy and trend fidelity despite differences in metric scales and variances—an evaluation protocol that is standard for performance prediction.

## 4 DATA CURATION

To evaluate at scale, we need a large number of diverse and realistic, open-ended pairs $(x_i, y_i)$. Rather than synthesizing experimental configurations and running automated experiments, which is both costly and unlikely to capture the kinds of problems practitioners actually investigate, we automatically curate experimental configurations and their reported results directly from the scientific literature. Our pipeline harvests redacted task descriptions and reported scores from published papers. We follow the pipeline in Figure 2: (i) collect arXiv experimental records (section 4.1); (ii) define the record provenance and paper types (section 4.2); (iii) recover each record's sources (section 4.3); (iv) extract a schema-aligned, redacted description (section 4.4); and (v) perform validation and quality control (section 4.5).

### 4.1 SOURCE CORPUS

We build on LLMEvalDB (Park et al., 2025), focusing on its 2023–2024 entries and 2025 entries from January to September, which are dominated by non-finetunable, proprietary models. This choice reduces variance from model/method tuning and better isolates dataset/task effects. Moreover, because current LLMs generally have a 2024 knowledge cutoff, restricting the 2025 split to post-cutoff papers mitigates leakage: the reported metrics could not have appeared in the training data. Each *experimental record* is a table cell with the dataset name, configuration context, and reported metric. Values are normalized to define $y_i$. We consider seven metrics with natural [0,100] scales: Accuracy, F1, Recall, Precision, ROUGE, BLEU, and EM.

### 4.2 RECORD PROVENANCE AND PAPER TYPES

Each record is grounded in two literature sources: the result paper, which reports the score, and the dataset paper, which introduced the dataset. These may coincide or differ. For example, Paper A introduces a QA dataset and reports accuracy for several models (result paper = dataset paper). Later, Paper B evaluates a new prompting/evaluation protocol on the same dataset under a revised setup (result paper $\neq$ dataset paper) (Park et al., 2021; Rangapur & Rangapur, 2024).

Figure 2: Overview of the PRECOG curation pipeline. For each record, we combine the result paper (Mitchell et al., 2023), which reports the score and specifies the evaluation protocol and configuration, with the dataset paper (Moskvichev et al., 2023), which provides the task description, to produce a schema-aligned input. The score 0.30 reported is normalized to serve as the target.

## 4.3 DATASET SOURCE PAPER RETRIEVAL

To compile faithful descriptions for each record, we first collect its sources. LLMEvalDB (Park et al., 2025) provides the result papers. When the result paper differs from the dataset paper, we automatically recover the latter by inferring the citation with an LLM, resolving it to an arXiv ID, downloading the PDF, and verifying the dataset name/title. We then run a rule-based check to confirm that the PDF explicitly mentions the dataset (string match).

## 4.4 EXTRACTING REDACTED INPUT DESCRIPTION

Given the sources for the result and dataset papers, an extractor LLM produces a schema-aligned, redacted description $x_i$ that summarizes the *task* (problem definition and input/label format), *data collection* (provenance, domain, collection protocol), *evaluation* (metric and protocol), *difficulty cues* (number of classes, long context, reasoning depth), the relevant *subset/split*, the *prompting strategy* (zero/few-shot, CoT), and any *other record-specific settings* (context-window, tool/retrieval use), when reported; see the Section E for illustrative examples.

When the retrieved sources do not actually cover the experimental record (e.g., due to retrieval errors from 4.3), we discard the case by having the extractor return `failed`, thereby correcting residual retrieval mistakes. For the remaining records, we remove dataset names, numerical results, and summary statistics to produce faithful, self-contained, anonymized.

## 4.5 DATA STATISTICS AND QUALITY CONTROL

We construct a bounded yet diverse corpus via stratified sampling over source papers, enforcing caps per paper and per dataset plus per-source quotas to prevent any single source from dominating. This yields 2,290 instances of records spanning distinct 1,019 dataset papers and 1,022 unique result papers, covering many datasets and configurations rather than a single benchmark (Ye et al., 2023). Because PRECOG is LLM-assisted, we further audited 30 random descriptions under a fixed protocol—checking anonymization (binary: no dataset names or performance numbers), schema coverage (5-point), and source support vs. the PDFs (5-point). Results (Table 2) show well-anonymized, accurate, and comprehensively grounded descriptions, suitable as schema-aligned, leakage-resistant inputs for text-only performance forecasting.

| Statistic | 2023–2024 | 2025 |
|---|---|---|
| **Record counts** | | |
| total instances | 767 | 1,523 |
| result paper = dataset paper | 504 | 549 |
| result paper ≠ dataset paper | 263 | 974 |
| **Paper counts** | | |
| unique result paper | 443 | 579 |
| unique dataset paper | 471 | 616 |
| result paper ∪ dataset paper | 631 | 956 |
| **Human validation (30-sample audit)** | | |
| Anonymous (0–1) | | 1.00 |
| Schema coverage (1–5) | | 4.61 |
| Source support (1–5) | | 4.87 |

Table 2: PRECOG statistics by collection period. Validation was conducted on 2023–2024 period.

## 5 EXPERIMENTS

**Baselines**   We compare against three families of baselines. First, we report the overall test-split mean across all metrics (Section 5.1) and the per-metric test-split means (Section 5.2). Because no training or validation set is available, the test-set average serves as a strong baseline for comparison. Second, we evaluate text-encoder–based regressors. We embed each description and train classical models—k-nearest neighbors (Guo et al., 2003) and XGBoost (Chen & Guestrin, 2016)—on these embeddings. With no dedicated training set, we use k-fold cross-validation (Fushiki, 2011): fit on k-1 folds, evaluate on the held-out fold, and average over folds. We set k = 10 and used E5-Mistral-7B (Wang et al., 2023), which is a strong dense embedding encoder baseline. Hyperparameters were tuned on a held-out validation split from the 2023–2024 version of PRECOG.

**LLMs**   We evaluate proprietary and open-source LLMs in both reasoning and non-reasoning modes to disentangle chain-of-thought effects from base generation quality. On the proprietary side, we use GPT-5 (OpenAI, 2025) in its reasoning configuration; on the open-source side, we include reasoning-enabled Qwen3-32B (Yang et al., 2025) and non-reasoning Qwen2.5-32B/72B (Team, 2024). For reasoning-capable models, we also run a no-reasoning variant (GPT-5 with `minimal` reasoning; Qwen3 with `no-thinking`) to quantify the contribution of explicit reasoning. We match prompts and decoding settings across modes so that performance differences can be attributed to reasoning. In all settings, the model produces a brief rationale and encloses its final estimate in `boxed{}`, which we parse in post-processing. GPT-5 uses default decoding hyperparameters, and Qwen3 uses the original authors' recommended settings.

### 5.1 EVALUATION RESULTS

We summarize the main validation results in Table 3. With the exception of GPT-5 (OpenAI, 2025), no baseline meaningfully outperforms the test-set global mean. GPT-5 achieves the lowest overall MAE (14.6), but still leaves substantial headroom. Performance is similar on the 2023–2024 and leakage-free 2025 splits, suggesting that prediction accuracy is not driven by training-data memorization. Moreover, reasoning ablations for GPT-5 and Qwen3 show that enabling reasoning mode improves performance prediction. Classical text-encoder baselines—kNN and XGBoost with k-fold CV—also fail to beat the global mean by a meaningful margin. Among Qwen variants (Team, 2024; Yang et al., 2025), Qwen3-32B in reasoning mode performs best, and is comparable to Qwen2.5-72B despite using roughly half as many parameters.

| Model | 2023–2024 | | 2025 | |
|---|---|---|---|---|
| | MAE $\downarrow$ | r $\uparrow$ | MAE $\downarrow$ | r $\uparrow$ |
| *Baselines* | | | | |
| Test set average | 21.4 | — | 23.0 | — |
| E5-Mistral + kNN | 19.9 | 0.30 | 21.2 | 0.22 |
| E5-Mistral + XGBoost | 20.3 | 0.30 | 21.4 | 0.20 |
| *LLMs* | | | | |
| GPT-5 | 14.7[†] | 0.69 | 14.6[†] | 0.62 |
| GPT-5 + *minimal-reasoning* | 15.0[†] | 0.68 | 15.5[†] | 0.52 |
| Qwen3-32B | 19.6[†] | 0.40 | 19.6[†] | 0.41 |
| Qwen3-32B + *no-thinking* | 22.9 | 0.37 | 22.5 | 0.37 |
| Qwen2.5-32B | 23.5 | 0.40 | 21.4 | 0.39 |
| Qwen2.5-72B | 22.4 | 0.44 | 19.5[†] | 0.43 |

Table 3: Prediction performance on PRECOG for 2023–2024 (767) and 2025 (1,523 instances). [†] indicates significance at $p < 0.05$ (one-sided binomial sign test) for the corresponding column.

### 5.2 PREDICTION HETEROGENEITY BY TASK, METRIC, TARGET, AND SETUP

We evaluate GPT-5 across seven metric groups by reporting per-metric MAE relative to a test-calibrated per-metric mean baseline, which is stronger than a single global mean. As shown in Figure 3, GPT-5 typically outperforms this baseline on both the 2023–2024 and 2025 splits. The ACCURACY subset—our largest slice—closely tracks the overall trend. In contrast, for Rouge and Precision, whose gold scores have low variance, LLMs sometimes fall below the baseline, underscoring the difficulty of forecasting such fine-grained metrics.

We analyze heterogeneity on the PRECOG 2025 snapshot along benchmark category, publication status, target model, and established datasets vs. new datasets in Table 4. We define *established*

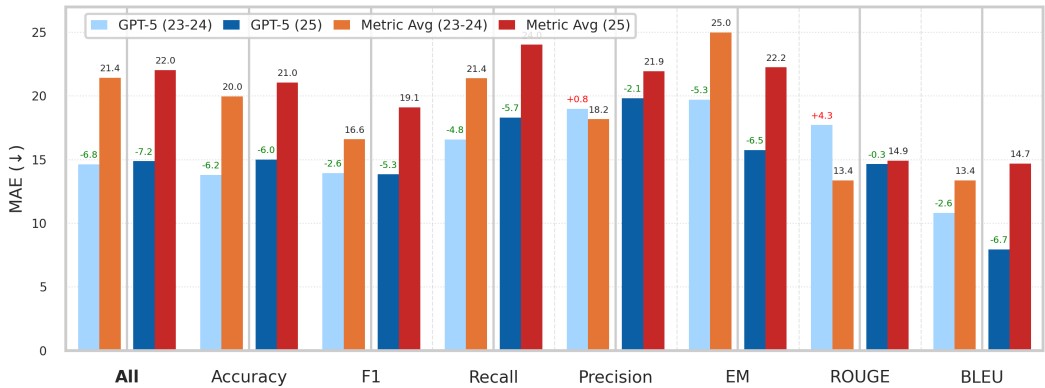

Figure 3: Per-metric MAE (↓) by model, grouped by evaluation metric and computed separately on the 2023–2024 and 2025 data subsets. The red and orange bars show the per-metric mean, labeled with its absolute MAE; the other bars show deviations from that mean within the corresponding year-split subset.

| Category | $n$ | GPT-5 | | Qwen3 | |
|---|---|---|---|---|---|
| | | MAE↓ | r↑ | MAE↓ | r↑ |
| Knowledge | 970 | 14.5 | 0.68 | 19.3 | 0.41 |
| Reasoning | 1100 | 14.8 | 0.67 | 19.5 | 0.37 |
| Math | 131 | 15.0 | 0.74 | 21.4 | 0.45 |
| Coding | 170 | 15.1 | 0.73 | 20.4 | 0.55 |
| Safety | 63 | 15.8 | 0.74 | 19.6 | 0.53 |
| IF | 382 | 15.5 | 0.66 | 20.7 | 0.40 |
| Multilingual | 60 | 12.0 | 0.78 | 17.9 | 0.60 |

(a) By benchmark category.

| Predictor | Test Set | $n$ | MAE↓ | r↑ |
|---|---|---|---|---|
| GPT-5 | Peer-reviewed | 486 | 15.4 | 0.69 |
| GPT-5 | Not peer-reviewed | 1036 | 14.5 | 0.68 |
| Qwen3 | Peer-reviewed | 487 | 20.7 | 0.40 |
| Qwen3 | Not peer-reviewed | 1036 | 19.1 | 0.42 |

(b) Peer-reviewed vs. non–peer-reviewed.

| Predictor | Target Model | $n$ | MAE↓ | r↑ |
|---|---|---|---|---|
| GPT-5 | GPT-4 | 406 | 13.9 | 0.73 |
| GPT-5 | GPT-4o | 1116 | 15.2 | 0.66 |
| Qwen3 | GPT-4 | 407 | 18.1 | 0.48 |
| Qwen3 | GPT-4o | 1116 | 20.2 | 0.40 |

(c) By target model (GPT-4 vs. GPT-4o).

| Predictor | Test Set | $n$ | MAE↓ | r↑ |
|---|---|---|---|---|
| GPT-5 | Established datasets | 973 | 14.7 | 0.69 |
| GPT-5 | New datasets | 549 | 15.1 | 0.67 |
| Qwen3 | Established datasets | 974 | 19.1 | 0.42 |
| Qwen3 | New datasets | 549 | 20.7 | 0.40 |

(d) Established datasets vs. New datasets.

Table 4: Subset ablation analyses on PRECOG 2025. Each panel reports MAE and Pearson **r** for GPT-5 and Qwen3-32B under different subset. Peer-reviewed entries are identified by matching result papers in DBLP. We define *established datasets* records as results on previously introduced datasets, and *new datasets* records as results reported in papers that introduce a new dataset.

*datasets* records as instances where the result reporting paper differs from the dataset paper, and *new datasets* records as instances where they are the same (the result paper introduced the dataset). Across categories, both GPT-5 and Qwen3-32B show similar patterns: multilingual benchmarks are easiest to forecast (lowest MAE, highest $r$), while math and instruction-following are harder, likely reflecting more diverse task formulations and evaluation setups. Performance is also very similar on peer-reviewed and non–peer-reviewed benchmarks, with slightly better MAE on non–peer-reviewed entries, suggesting that venue and "maturity" have limited impact.

Conditioning on the target model, both predictors perform better on GPT-4 than GPT-4o, indicating that newer or more variable targets are modestly harder to forecast, plausibly because GPT-4 results were more exposed during LLM training. Finally, MAE and correlation are nearly identical for established datasets vs. new datasets records, implying that the predictors transfer well from established datasets to newly introduced, previously unseen datasets.

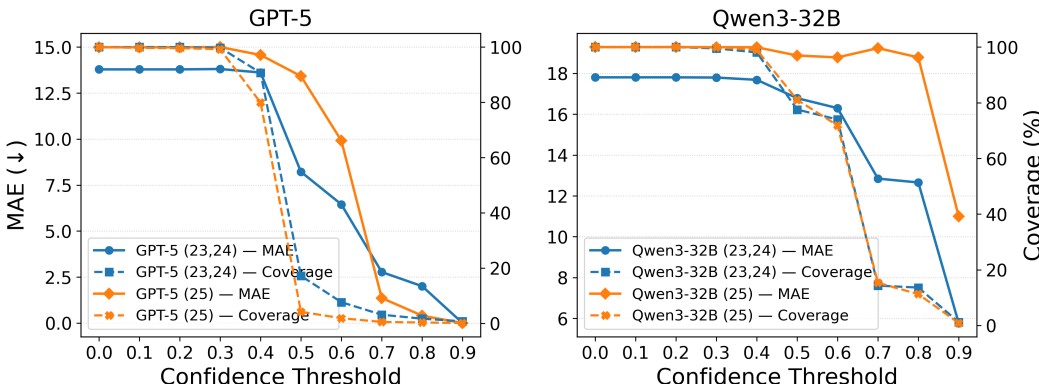

Figure 4: MAE (left axis) and coverage (right axis) as a function of confidence thresholds (scores ≥ band lower bound) for GPT-5 and Qwen3-32B across different year splits (2023-2024 vs 2025). Higher thresholds retain fewer examples (lower coverage) and generally yield lower MAE. Confidence values are derived from a 10-bin categorical output produced by the verbalized confidence calibration.

### 5.3 HUMAN PREDICTION RESULTS

In the previous sections, we demonstrated that LLMs can generate accurate predictions from dataset descriptions. A natural next question is how these predictions compare to those of human experts, who are known to struggle with probability judgments and decisions under uncertainty (Kahneman & Tversky, 1972). To explore this, we asked six graduate students and one professor of computer science, all specializing in NLP and machine learning, to each predict 20 accuracies based solely on the same text descriptions provided to models.

| Model | MAE | r |
|---|---|---|
| Human | 19.6 | 0.40 |
| Test subset average | 21.7 | – |
| GPT-5 | 13.6 | 0.69 |

Table 5: Human Prediction Results on Accuracy subset ($n = 140$).

On this 140-item subset, human predictions correlated moderately with ground truth ($r = 0.403$) and outperformed the mean baseline in terms of MAE (19.6 vs. 21.7). Nevertheless, GPT-5 achieved substantially stronger results, indicating that while humans can do the task reasonably well, advanced reasoning models achieve markedly lower MAE. Although the study has a small sample size, the results primarily serve to illustrate the inherent difficulty of the task. While human forecasters might improve with additional training (Tetlock et al., 2014), the fact that LLMs already outperform researchers on this performance prediction task suggests that benchmark prediction models have the potential to provide meaningful support to practitioners when planning experiments.

## 6 CONFIDENCE CALIBRATION

Some experimental outcomes may be inherently difficult to predict, raising the question of whether models can reliably estimate their own prediction accuracy. To examine this, we elicit a scalar, self-reported confidence for each prediction (following Yoon et al. (2025)) and evaluate calibration by filtering to items with confidence ≥ $\tau$. For each $\tau$, we report the MAE on the retained subset together with its coverage. This gives a simple view of the error–coverage trade-off. We report confidence curves for the accuracy subset, which dominates PRECOG.

Figure 4 shows results on the accuracy subset for GPT-5 and Qwen3-32B on different year-subset split. As the confidence threshold increases, coverage declines and MAE tends to decrease, indicating that higher reported confidence is, on average, associated with more accurate forecasts. The 2023–2024 split tends to be better calibrated—MAE drops more sharply as the confidence threshold increases—which may reflect that the evaluated LLM has seen more similar examples from this period during training. Overall, these curves suggest reasonable alignment between self-reported confidence and error without making strong claims about perfect calibration.

## 7 RETRIEVAL ANALYSIS

We assess how retrieval contributes to performance prediction using tool-call counts and query token diversity (tokenized with GPT-4o (Hurst et al., 2024)). All models share a custom retriever over the arXiv API,[1] which returns canonical arXiv PDF links. To avoid leakage, we exclude each record's source papers by identifier during retrieval. We initially tried GPT-5's built-in web search—which is more token-efficient but corpus-uncontrolled—and found that it frequently surfaced source papers: in an audit, roughly 70% of evaluation source papers became identifiable, so we instead use the custom retriever for GPT-5 despite its higher cost. We report results with retrieval enabled and disabled to measure gains from literature grounding beyond the model's prior. Each query returns at most two documents, and the agent may issue up to four retrievals under ReAct-style prompting (Yao et al., 2023).

| | GPT-5 | Qwen3-32B |
|---|---|---|
| *Prediction performance* (**MAE / r**) | | |
| No *Search* | 14.7 / 0.69 | 19.6 / 0.40 |
| With *Search* | 14.0 / 0.73 | 20.8 / 0.31 |
| *Search-tool usage & token diversity* | | |
| Avg. search calls | 3.06 | 0.89 |
| Std. search calls | 0.87 | 0.32 |
| Unique tokens (all) | 2,140 | 794 |
| Unique tokens (random) | 1,321 | — |

Table 6: Search-tool usage, token diversity, and prediction performance for GPT-5 and Qwen3-32B. Token counts are measured with the GPT-4o tokenizer. (all) aggregates all formulated queries; (random) uses one random query per instance.

As shown in Table 6, search improves GPT-5's performance prediction but not Qwen3's. GPT-5 issues more—and more variable—search calls per example than Qwen3-32B, and its queries are more lexically diverse, suggesting it aggregates evidence across a broader set of retrieved documents, whereas Qwen's lighter tool use limits potential gains. Yet when we asked an LLM to rate retrieved documents on a 5-point relevance scale, most were only "moderately relevant": the retriever often surfaces high-level related work rather than papers that add much beyond pretraining and our structured descriptions. This helps explain why sparse retrieval has limited impact and can even hurt Qwen3 in some cases. Overall, retrieval is not the main source of forecasting gains in PRECOG; most signal comes from reasoning over curated descriptions plus the model's own knowledge.

## 8 STREAMING PREDICTION

Using PRECOG, GPT-5 outperforms the per-metric test-set mean baseline. Nevertheless, residual answer leakage cannot be entirely excluded—for instance, when subsequent papers copy baseline results from the original publication. Moreover, we disabled built-in web search because it risks leakage, for example by retrieving the target papers during prediction. To further validate our experimental setup under more realistic conditions, we adopt a *streaming prediction* setting, which tests models on their ability to predict performance on newly reported arXiv results immediately upon release, before they can be indexed by search engines or incorporated into training.

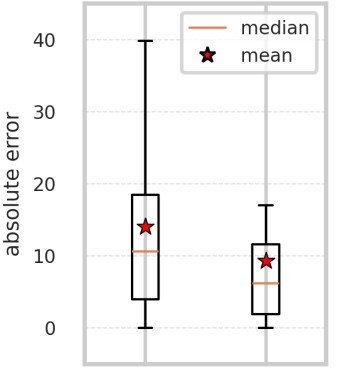

Figure 5: Streaming prediction error boxplots of GPT-5+Web-Search.

**Setup** We collect candidates from arXiv's official interfaces (API/RSS) for `cs.CL`, `cs.LG`, and `cs.AI`, and fetch PDFs via the recommended hosts, respecting `robots.txt` and published rate limits; we also notified arXiv staff in advance (see Section G).

A daily pipeline then (i) identifies papers of interest, (ii) extracts table records, (iii) builds redacted descriptions, and (iv) issues predictions—end-to-end before third-party search engines index the papers. In this setting, we restrict to records whose metrics map to our seven metric groups and expand the model axis to the GPT family. We permit built-in web-search here, as leakage from prior literature is negligible for freshly released papers. We collected 52 papers containing 79 previously

---

[1] https://info.arxiv.org/help/api/index.html

unseen experimental records and datasets between 2025-09-05 and 2025-09-23. During search, GPT's web tool may surface the target paper. We match arXiv IDs and verify content to detect any such access, and we exclude those predictions to prevent leakage.

**Results** Table 7 summarizes streaming performance for GPT-5 (+*Search*). On newly released datasets and unseen experiment–dataset combinations, the model achieves accuracy comparable to its performance on PRECOG (MAE = 14.0). The box plot of absolute errors (Figure 5) compares all predictions with the high-confidence subset (>50% confidence, 23 samples), showing a visibly tighter error distribution for the latter (avg. 9.3). Finally, in a self-referential test that predicted the performance of the task described in **this paper** (not yet on

| Model | MAE | r |
|---|---|---|
| Test set average | 25.5 | — |
| GPT-5 | 15.0 | 0.74 |
| GPT-5 + Search | 14.0 | 0.78 |

Table 7: Streaming environment prediction results on 79 samples.

arXiv), GPT-5+Search predicted its own performance at MAE 16.0 versus the realized 14.7 from Table 3. These results indicate that forecasting performance based solely on text-only descriptions could be a practical and achievable approach in real-world applications.

# 9 CONCLUSION & DISCUSSION

**Conclusion.** We recast evaluation as *forecasting* and introduce PRECOG, a leakage-mitigating corpus of redacted, schema-aligned description–performance pairs. GPT-5 produces moderately accurate forecasts with calibrated uncertainty, and remains effective in a streaming setting where results are predicted for newly released papers prior to broad indexing. These findings indicate that anticipatory evaluation is feasible and practically useful for estimating task difficulty, prioritizing experiments, and reducing compute.

**Discussion: scope and limitations.** Our goal is not to replace full benchmark evaluations. The forecaster is a pilot-stage decision aid: it helps estimate task difficulty for new datasets or setups, narrow large design spaces to a smaller set of promising configurations, and plan pilot studies and resource allocation. Its granularity is insufficient for leaderboard-level distinctions (e.g., 73.2% vs. 74.0%), but is informative for separating clearly weak regimes (e.g., 40∼ 50) from potentially deployment-ready regimes (e.g., ∼ 80). Also, additional limitation include dependence on the quality and specificity of textual descriptions and potential biases inherited from the training distribution. Despite our leakage controls (post-cutoff, streaming), residual leakage risks remain and should be monitored.

**Discussion: cost and latency.** Using GPT-5 (medium reasoning, no search) via the Batch API costs about 0.01 per query and averages ∼132 seconds per prediction with high parallelism. For open-source baselines (e.g., Qwen), we run single-GPU inference on an A40 with a maximum generation length of 32,768 tokens; a single prediction averages ∼48 seconds. While not directly comparable to end-to-end benchmark runs, description-based forecasting is markedly more cost- and time-efficient, making it well suited for early screening and study design.

**Discussion: limits of current open-source models.** Despite progress, open-source baselines lag behind GPT-5 in both accuracy and calibration. They show brittle tool use (inconsistent gains from retrieval), weaker representations of fine-grained cues in long descriptions, and higher variance across metric families (e.g., small-variance metrics like BLEU/ROUGE). Improved reasoning and uncertainty modeling are needed for reliable deployment.

**Outlook: toward specialized forecasting models.** To accelerate scientific discovery, we advocate purpose-built regressors that leverage LLM reasoning. In particular, **reasoning-aware regression** trains a regression model on top of reasoning LLMs, while **better use of retrieval tools** focuses on learned query planning and selective evidence use. An important future direction is to make these forecasting systems robust for *new* models with very few or no prior evaluations, for example by incorporating stronger priors, actively collecting a small set of seed evaluations, or using richer hybrid text–metadata representations. Developing these components on top of PRECOG and the streaming benchmark should yield more accurate, trustworthy, and cost-efficient forecasting systems.

## 10    ETHICS STATEMENT

We rely exclusively on publicly available research artifacts (e.g., arXiv PDFs and published results); no human subjects or personally identifiable data are involved, and all usage follows licenses and provider terms. To reduce leakage, we redact dataset identities and block retrieval of each record's table-source and dataset-source papers; because residual cues may persist, we also evaluate a time-gated, zero-leakage setting on genuinely unseen datasets. Focusing on arXiv means some records lack full experimental details, which can introduce label noise; to aid auditing, we release venue metadata per record and document known issues. Our strongest results use large proprietary models, raising equity and environmental concerns; we therefore report compute, provide open baselines, and encourage reproduction with smaller models. Finally, the corpus is predominantly English, limiting generality; expanding multilingual coverage is an important direction for future work.

## 11    REPRODUCIBILITY STATEMENT

We will release all artifacts needed for exact or statistically equivalent reproduction: the full PRE-COG curation stack (scripts, leakage filters, redaction utilities), the forecasting pipelines (retrieval, prompting, decoding, scoring), and the evaluation scripts. Our retriever queries the arXiv API, returning at most two PDFs per query and up to four queries per instance (ReAct-style); we will also release the exact query templates and our custom arXiv wrapper. To model unseen datasets, we enforce a *zero-leakage* protocol by time-gating retrieval to snapshots before each dataset's index date, and we publish per-record snapshot dates plus a script. All prompts (system/instruction/user and ReAct) are released verbatim; decoding parameters are fixed and logged, defaulting to greedy where configurable. For GPT-5, decoding settings are provider-controlled; we use defaults and record access dates. Baselines (E5-Mistral + kNN/XGBoost; Qwen3 variants) are used off-the-shelf; we provide exact checkpoints or API identifiers and access dates, and no baseline is finetuned. Upon acceptance, we will release code, data, prompts, configs, and logs under `<MIT LICENSE>`; non-redistributable third-party assets are referenced with exact versions and retrieval instructions.

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

## A  DETAILS OF DATA CURATION

This section provides data-curation details not covered in Section 4.

**LLMs**  For curation we used GPT-5-mini (OpenAI, 2025). The model extracted schema-aligned descriptions from source papers and handled dataset-source retrieval annotations. These choices were based on a pilot run and seed-result quality.

**Dataset-source arXiv annotation**  Given the result paper PDF, we prompted GPT-5-mini to infer the title of the dataset-source paper, queried the arXiv API to resolve an identifier, and downloaded the PDF. We improve precision by automatically checking content (e.g., verifying that the dataset name appears in the retrieved paper).

**Description extraction**  We excluded appendix tables during extraction. To disambiguate records, we supplied the pre-normalization metric value as context. For efficiency, all PDFs were truncated to the first 12 pages during retrieval and extraction.

**Additional filtering**  All records involve GPT-4 (Achiam et al., 2023) or GPT-4o (Hurst et al., 2024), fixing the model/method axis to focus on dataset/task variation. We excluded multimodal domains, correlation-only metrics (not on a 0–100 scale), and overall benchmark aggregates (e.g., MMLU) via heuristic filters.

## B  ANNOTATION GUIDELINES

### B.1  DESCRIPTION QUALITY

This section details the human annotation protocol used to evaluate the quality of redacted, schema-aligned descriptions against their source PDFs. Annotators were instructed to verify that each description is anonymized, complete where possible, and strictly grounded in the provided sources.

**Annotation Task**  Annotators were given:

- The extracted input description for a given experimental record.
- The corresponding result and dataset paper PDFs.

The task was to check whether the description faithfully reflects information from the PDFs, is anonymized, and adheres to the fixed schema.

### B.2  EVALUATION CRITERIA

Three dimensions of quality were assessed:

**Anonymization (Binary).**  Descriptions must not include dataset or subset names, and must exclude performance numbers drawn from the main paper. Annotators assign either **Accept** (score: 1.0) or **Reject** (score: 0.0).

**Schema coverage (1–5).**  Checks whether all schema fields are filled appropriately when information is available in the PDFs, and "None" is used only when the information is truly absent.

- **Score 1**: Ignores the schema; most fields missing or collapsed.
- **Score 2**: Very few fields filled; many omissions; "None" often misused.
- **Score 3**: About half of fields filled; some omissions or misplacements.
- **Score 4**: Nearly all fields correctly covered; rare omissions or minor misplacements.
- **Score 5**: Complete and consistent coverage; "None" only when truly absent; perfectly aligned with schema.

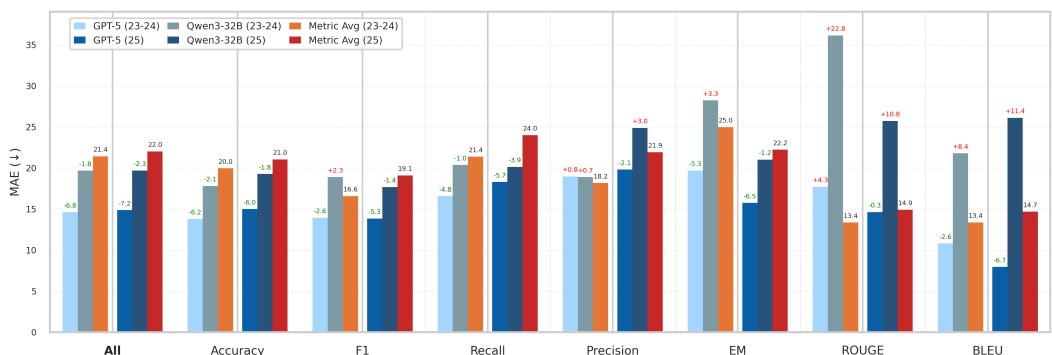

Figure 6: Per-metric MAE (↓) for all baselines and LLMs, grouped by evaluation metric (e.g., Accuracy, Exact Match (EM), Recall, F1) and computed separately on the 2023–2024 and 2025 data subsets. The red and orange bars show the per-metric averages for each year split, with their absolute MAE values labeled, and the remaining bars indicate how much each model deviates from the corresponding yearly average.

**Factual grounding (1–5).** Ensures every statement in the description is directly supported by the PDFs, without speculation or outside knowledge.

- **Score 1**: Mostly unsupported or invented; contradictions with PDFs.

- **Score 2**: Frequent unsupported claims; weak sourcing.

- **Score 3**: Generally grounded, but several unsupported claims or mild over-interpretation.

- **Score 4**: Fully grounded with only minor overstatements; no contradictions.

- **Score 5**: Entirely faithful; precise paraphrases of the PDFs; every fact traceable.

## B.3 HUMAN PERFORMANCE PREDICTION

This subsection specifies the human annotation protocol for predicting task performance from redacted, schema-aligned descriptions. Annotators produce a single accuracy estimate and a brief rationale *without consulting external sources*, enabling an assessment of human forecasting difficulty and behavior under text-only constraints.

**Annotation Task**   Annotators are given:

- A redacted description of an experimental setup (model, dataset/task, evaluation protocol, and salient configuration details).

- A fixed-time budget per item and an interactive prompt that records their identity (first name) and assigns a shard automatically.

For each item, annotators must output:

- **Predicted accuracy** (integer in 1–100).

- **Rationale** (1–3 sentences) briefly justifying the estimate.

**Instructions and Constraints**

- **No external search.** Rely solely on the provided description and personal prior knowledge; do not use web search, papers, or tools.

- **Timing.** Aim to complete each prediction within 2–3 minutes.

- **Batch size.** Each annotator completes 20 samples per session.

| Model | MAE | Pearson Correlation |
|---|---|---|
| E5-Mistral-7B + KNN | 21.2 | 0.22 |
| E5-Mistral-7B + XGBoost | 21.4 | 0.20 |
| Qwen3-8B-Embed + KNN | 20.8 | 0.27 |
| Qwen3-8B-Embed + XGBoost | 21.6 | 0.21 |
| KaLM-Embedding-Gemma3-12B-2511 + KNN | 20.8 | 0.26 |
| KaLM-Embedding-Gemma3-12B-2511 + XGBoost | 21.5 | 0.21 |

Table 8: Results on the PRECOG-2025 dataset with different embedding backbones and regressors. Newer, stronger embedding models provide modest gains, but improvements are relatively small compared to the overall bottleneck from compressing each model–task pair into a fixed-size vector.

## C  SUPPLEMENTARY EXPERIMENTAL RESULTS

**Per Metric MAE for all the baselines & LLMs**  We report per-metric MAE for all baselines and LLMs, along with a test-calibrated per-metric mean baseline across seven metric groups in Figure 6.

**Additional results with newer embedding backbones.**  We ran additional experiments on the PRECOG-2025 dataset using more recent embedding models that are competitive or state-of-the-art on the MTEB leaderboard, including QWEN3-8B-EMBED (Zhang et al., 2025) and KALM-EMBEDDING-GEMMA3-12B-2511 (Hu et al., 2025), combined with both KNN and XGBoost regressors. The results are summarized in Table 8. As the table shows, newer and stronger embedding models yield modest improvements in MAE and correlation, but the gains are relatively small.

## D  ILLUSTRATIVE CASE STUDY EXAMPLE

### D.1  PUZZLE QA CASE STUDY (ON-AIR VS. OFF-AIR PUZZLES).

We consider a benchmark of verbal reasoning puzzles drawn from a long-running radio show Wu et al. (2025). The original authors distinguish two settings: (i) on-air puzzles, which are short, pattern-based word questions intended to be solved live in a few seconds by callers during the broadcast, and (ii) off-air weekly challenges, which are substantially harder puzzles that listeners may think about for days before submitting an answer. In the released benchmark, GPT-4o achieves around 65% accuracy on the easier on-air subset, but only about 6% on the more challenging off-air subset.

To simulate how our system could assist benchmark design *before* either subset is fully constructed, we ignore these empirical numbers and assume a designer only has two candidate formulations in mind. The designer writes two short, detailed descriptions: an on-air description emphasizing that puzzles are designed for live radio, solvable in seconds, with strongly structured patterns and relatively shallow wordplay; and an off-air description emphasizing

| Subset | Gold / Pred. acc. (%) | Abs. error (pp) |
|---|---|---|
| On-air | 65 / 68 | 3 |
| Off-air | 6 / 14 | 8 |
| Overall | — | 5.5 (MAE) |

Table 9: Gold and predicted GPT-4o accuracies on the on-air and off-air puzzle subsets, along with absolute error in percentage points (pp).

that puzzles are intended as week-long challenges, often involving multi-step verbal reasoning, broader general knowledge, and a much larger search space over possible candidate answers. These natural-language descriptions, together with the target model name, are provided as the only inputs to our predictor; crucially, no performance statistics from the original benchmark are revealed.

Under this purely description-based setup, our system predicts 68% accuracy for GPT-4o on the on-air variant and 14% on the off-air variant as in Table 9. When compared to the eventual empirical scores (65% and 6%), the corresponding absolute errors are 3 and 8 percentage points, respectively, yielding a mean absolute error of 5.5 percentage points. The model correctly predicts a large drop in performance going from on-air to off-air.

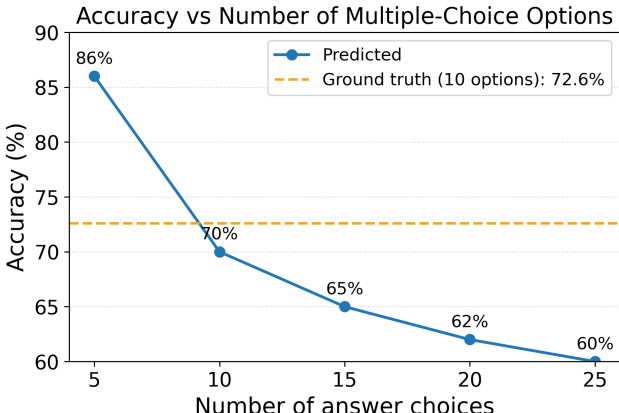

Figure 7: Predicted GPT–4o accuracy as a function of the number of multiple-choice options for a multitask reasoning benchmark. The solid line shows our description-based predictions for 5, 10, 15, 20, and 25 options, and the dashed horizontal line marks the reported ground-truth accuracy of 72.6% for the 10-option configuration used in practice.

From a benchmark designer's perspective, this ex-ante signal is already actionable. Suppose the designer's goal is to avoid both saturation (a task that is too easy for current models) and degeneracy (a task on which models essentially always fail). The predicted 68% versus 14% suggests that the on-air formulation is likely to be quickly saturated by strong models, whereas the off-air formulation is unlikely to be impossible (for example, 0% accuracy), and instead lies in a more interesting, mid-range difficulty regime. In a hypothetical setting where neither dataset yet exists, the designer could draft natural-language specifications for an "easy/live" and a "hard/take-home" variant, query our system once to obtain rough performance predictions on each, and then use these predictions to prioritize constructing and cleaning the more promising hard variant (off-air) first. Only that variant would need to be validated initially with empirical model runs, reducing the need to fully curate both candidate datasets and run pilot experiments on each. While our predictions still require downstream validation, they provide a quantitative shortcut that can guide early benchmark design decisions, turning vague intuitions ("off-air seems harder") into concrete, model-specific expectations.

### D.2 MMLU-PRO: CHOOSING THE NUMBER OF ANSWER OPTIONS

To illustrate how description-based prediction can support design-time decisions, we consider a multitask multiple-choice benchmark in the style of MMLU-Pro (Wang et al., 2024), and focus on a single design configuration: the number of answer options per question. We imagine a benchmark designer who has not yet built the dataset and is deciding between 5, 10, 15, 20, or 25 options. For each configuration, the designer writes a natural language description (same questions and domains, different number of options) and queries our predictor, which estimates GPT–4o (Hurst et al., 2024) accuracy from text alone, without access to any items.

Figure 7 summarize the results. Using only these textual descriptions, our predictor (GPT–5) estimates GPT–4o accuracies of 86%, 70%, 65%, 62%, and 60% for 5, 10, 15, 20, and 25 options, respectively, while the reported ground truth accuracy at 10 options is 72.6% from the original paper (Wang et al., 2024). The predictions are monotone in the expected direction and reasonably calibrated at the configuration used in practice.

This example shows how a single batch of zero-example predictions can already guide a practical choice. Before collecting any data, the designer can (i) rule out clearly saturated settings such as 5 options, (ii) avoid very difficult settings such as 20–25 options that risk making the benchmark unusable for current models, and (iii) focus annotation and compute on intermediate configurations around 10–15 options where performance is expected to fall in a useful range. In this sense, our method provides a coarse but actionable quantitative signal for prioritization and resource allocation at design time.

## D.3 ON-AIR SETTING DESCRIPTION

**Model:** GPT-4o

- **Task:**
  - Open-ended answer generation for short word puzzles presented in a structured format.
  - Input: a shared natural-language description that specifies a wordplay pattern or rule (e.g., a relationship between two words), plus one concrete question that instantiates this pattern.
  - Answer space: unconstrained natural-language strings, typically a single word or very short phrase (for example, a word that becomes a synonym of another after a letter rearrangement).
  - Evaluation target: determine whether the model's generated text contains the correct target word or phrase that satisfies the described constraint for that question.

- **Data collection:**
  - Derived from a recurring live puzzle segment on a radio program (on-air).
  - In each segment, a host first reads a brief description that defines a wordplay pattern, then poses a sequence of individual questions, each consisting of concrete words or phrases that fit the pattern; callers attempt to solve these in real time during the broadcast.
  - For each description, multiple question–answer pairs are associated with the same underlying pattern, yielding clusters of structurally similar items.
  - These on-air questions and their solutions were previously collected and standardized into a benchmark by earlier work, and are reused here to evaluate modern models; the present setup relies on that standardized corpus rather than reconstructing the scraping and transcription pipeline.

- **Evaluation:**
  - Metric: accuracy over individual question–answer pairs.
  - Each on-air question is treated as one evaluation item; the model is prompted with the description and the corresponding concrete question, and produces a free-form completion.
  - A completion is counted as correct if it contains the gold solution string for that question, ignoring superficial differences such as capitalization and punctuation.
  - No partial credit is awarded: items are marked simply correct or incorrect, and the reported score is the mean accuracy over all on-air questions in this subset.

- **Difficulty characteristics:**
  - Puzzles are explicitly designed to be solvable within a few seconds by human callers during a live broadcast, making them easier than the show's separate, week-long take-home challenges.
  - The shared description typically makes the pattern very explicit (for example, specifying exactly how two given words must be related), and the associated questions closely follow this pattern, providing strong guidance about the required transformation or relationship.
  - Most items involve letter-level or word-level constraints (such as rearranging letters to obtain a synonym), sometimes combined with simple semantic relations (e.g., basic synonymy), rather than deep multi-step reasoning.
  - Because the same description applies to multiple questions, the underlying reasoning structure is repeated within small clusters; once the model infers the pattern from the description, each additional question in that cluster is another instance of the same structure, reducing overall difficulty relative to more heterogeneous, one-off puzzles.

- **Subset:**
  - This record corresponds specifically to the on-air subset of puzzles from the radio program.

- Every item in this subset is derived from questions intended to be answered live during the broadcast under tight time constraints.
- The subset excludes the separate weekly take-home challenges and does not apply further filtering by puzzle type or knowledge category beyond what was done in the original benchmark construction.

- **Prompting strategy:**
  - Zero-shot generation on the on-air puzzles.
  - For each question–answer pair, the prompt consists of the textual description of the pattern together with the specific instantiation (the question containing the concrete words or phrases).
  - No in-context examples from other puzzles, no few-shot demonstrations, and no explicit chain-of-thought exemplars are provided.
  - The model is allowed to respond in free-form natural language; it may include reasoning text and then an answer, or simply output an answer, and there is no enforced answer template.
  - A single sample is generated per question; there is no best-of-$n$ or multi-sample aggregation for this record.

- **Other settings:**
  - The system is used as a general-purpose large language model in its standard, non–reasoning-augmented mode, without special test-time "extended thinking" features or dedicated reasoning tokens.
  - Decoding employs a moderate amount of stochasticity (nonzero temperature with nucleus sampling), consistent with settings used for non-reasoning baselines on related tasks.
  - No retrieval augmentation, external tools, or auxiliary context sources (such as web search or separate knowledge bases) are used; the only information provided at inference time is the puzzle description and the question text.
  - Context length requirements are minimal because each on-air item is short, and no special truncation rules are applied beyond the model's default maximum output length.

**Target metric**: 65 (accuracy)

## D.4 OFF-AIR SETTING DESCRIPTION

**Model:** GPT-4o

- **Task:**
  - Open-ended answer generation for verbal puzzles.
  - Input: a short natural-language puzzle description specifying a pattern or constraint over words, letters, sounds, or common-knowledge facts.
  - Answer space: effectively all possible strings in the language; solutions are typically a single word or short phrase, but can sometimes involve multiple words or multiple solution strings separated by punctuation.
  - Evaluation target: check whether the model's generated text contains the correct solution phrase(s) that satisfy the puzzle constraints.

- **Data collection:**
  - Derived from transcripts of a long-running weekly radio puzzle segment.
  - Each transcript includes the previous week's challenge puzzle and its official solution; these were scraped from publicly available web pages and then manually cleaned.
  - Cleaning operations include making implicit temporal or geographic references explicit (for example, replacing relative time expressions like "last year" with the corresponding calendar year, or clarifying that a greeting comes from a country other than a reference country).
  - Puzzles whose solutions admit very large numbers of valid answers (for example, cases where listeners submitted many distinct correct solutions) are removed because they are not well suited to automatic evaluation.
  - Verbal explanations are stripped from the stored solutions so that the gold answer consists only of the target word(s), rather than a full explanatory sentence.

- **Evaluation:**
  - Metric: accuracy over individual puzzles.
  - For each puzzle, the model is run once and produces a free-form completion that may include both reasoning and a final answer.
  - Capitalization and punctuation are ignored during scoring; an output is judged correct if every phrase in the stored gold answer appears somewhere in the generated text.
  - If the gold solution consists of multiple phrases, all of them must be present; otherwise the item is marked incorrect.
  - No partial credit or ranking is used; each puzzle is counted as correct or incorrect, and mean accuracy over the full set of weekly challenges is reported.

- **Difficulty characteristics:**
  - Puzzles are designed for adult native speakers with broad general knowledge, and are intended to be sufficiently challenging that many listeners may think about them for days before the answer is revealed.
  - Although they are hard to solve, solutions are straightforward for humans to verify once known, which also makes model errors easy to inspect.
  - Many puzzles involve nontrivial wordplay such as letter rearrangements, constraints on letter counts, or sound-based relationships (rhymes, homophones), often combined with semantic or world-knowledge constraints (for example, cities, foods, well-known people, greetings, or other culturally specific entities).
  - The collection spans multiple verbal-reasoning types, including letter-level manipulations, sound-based reasoning, visual or positional layouts of letters, riddles, and related trick-based formulations.
  - Knowledge domains are broad, covering general knowledge, geography, history, popular culture, sports, politics, and arts; individual puzzles often require multi-step reasoning and candidate search rather than direct retrieval of a single fact.

- **Subset:**

- This record corresponds to evaluation on the full collection of cleaned weekly challenge puzzles.
- No additional restrictions are applied by puzzle type, knowledge category, or time period beyond the cleaning and filtering described above.

- **Prompting strategy:**
  - Zero-shot generation on the weekly challenge puzzles.
  - For each puzzle, the input to the model consists solely of the puzzle text itself; there are no added instructions about answer format, no worked examples, and no few-shot demonstrations.
  - The model is free to produce both a reasoning process (if any) and a final answer in an unconstrained natural-language completion.
  - There is no explicit chain-of-thought prompting, and no special answer template is imposed; the system is treated as a standard text-completion model.
  - Each reported accuracy is based on a single completion per puzzle, without best-of-$n$ sampling or aggregation.

- **Other settings:**
  - The evaluated model is a non-reasoning variant within a broader model family that also includes reasoning-augmented versions.
  - It does not use extended test-time "thinking" modes or dedicated reasoning tokens; instead, it is run in its default completion mode.
  - Decoding uses a relatively low sampling temperature and a nucleus-sampling cutoff (top-$p$), matching the configuration adopted for non-reasoning baselines on this benchmark.
  - No external tools, retrieval components, or auxiliary context sources beyond the puzzle text are employed.
  - Output length is not specially tuned beyond the default limits for this model class, and no special truncation rules are applied; each puzzle is handled as a single, short prompt with its corresponding single completion.

**Target metric**: 6 (accuracy)

# E  PRECOG SAMPLES

We randomly sample 5 instances from PRECOG for the demonstration.

**Source Papers**: result paper=`2306.16638`

**Model**: GPT-4

**Description**

- **Task:**
  - Binary classification of sentence pairs.
  - Input: two short natural-language sentences presented as a pair.
  - Label space: two classes indicating whether the second sentence logically negates the first (Yes/No; encoded as a binary output).
  - Evaluation target: identify logical negation between sentences rather than relying on surface-level cues.
- **Data collection:**
  - Custom corpus created by the author using designed sentence templates.
  - For each pair, components were systematically negated (or left unmodified) to generate balanced positive and negative examples.
  - Construction followed classical logic principles to ensure "negation" reflects logical contradiction, with emphasis on structures prone to errors (e.g., conditionals).
  - Sentence length: approximately 5–20 words, with a roughly normal distribution.
  - Domains: diverse everyday and professional contexts; pairs span multiple topics.
- **Evaluation:**
  - Metric: Recall calculated for the positive class (pairs where the second sentence negates the first), comparing model predictions with gold-standard labels across the entire dataset.
- **Difficulty characteristics:**
  - Balanced two-class classification task.
  - Inclusion of logically complex constructions (e.g., conditional statements), rendering surface-level "not" cues unreliable.
  - Short but variable sentence lengths (5–20 words).
  - Designed to assess logical reasoning robustness across diverse contexts.
- **Prompting strategy:**
  - Zero-shot classification.
  - A fixed instruction presents the two sentences and asks whether the second negates the first, requiring a binary response (Yes/No encoded as 1/0).
  - No in-context examples, chain-of-thought reasoning, or auxiliary decoding constraints used.
- **Other settings:**
  - Evaluated via an API for the later-generation model in this family; each pair assessed once using the fixed prompt.
  - No retrieval augmentation, external tools, or special truncation/windowing rules reported.

**Target metric**: 72.8 (recall)

**Source Papers**: result paper=`2403.00092`

**Model**: GPT-4

**Description**

- **Task:**
  - Predict symbolic planning domain actions from natural-language procedural text.
  - Input: a natural-language procedure together with a domain header listing available types, predicates, and the names of actions to be filled in.
  - Output: a domain file's action definitions; each action must include typed parameters, a conjunction of preconditions, and a conjunction of effects, expressed in a formal planning-language syntax.
  - Constraint: models must not add, remove, or rename actions.
  - Evaluation target: reproduce the gold action definitions so that the predicted domain file matches the reference.
- **Data collection:**
  - Source texts are open-domain how-to guides from the web covering varied everyday and survival-style topics.
  - For each procedure, trained experts manually created paired symbolic planning representations: a domain file (types, predicates, actions) and associated problem files (entities, initial and goal states).
  - Representations are planner-ready and not tied to any specific simulator.
- **Evaluation:**
  - Metric: action-level accuracy under intrinsic evaluation.
  - Protocol: compare predicted actions with the reference actions in the gold domain file using a semantic-equivalence check that disregards variable renaming and conjunct order; parameters must match by type, and predicates must align in arity and argument mapping.
- **Subset:**
  - intrinsic—only direct semantic alignment of predicted action definitions is assessed; no plan execution/solvability is considered.
- **Difficulty characteristics:**
  - Open-domain content with diverse topics.
  - Requires extracting relevant information, inferring implicit entity states, and precisely translating into a low-resource, domain-specific formal language.
  - Predicting preconditions is generally more challenging than predicting effects.
  - Input texts can be long (from concise summaries to multi-paragraph procedures), creating context-length and information-extraction challenges.
- **Prompting strategy:**
  - Zero-shot direct instruction without step-by-step reasoning scaffolding.
  - The prompt provides the domain header (types, predicates, action names) and the relevant procedure text, and asks the model to complete each action's parameters, preconditions, and effects in the required formal syntax while preserving the given action set.
- **Other settings:**
  - Evaluated via an API for GPT-4; the input budget was capped at roughly ten thousand tokens for these runs.

**Target metric**: 15.9 (accuracy)

**Source Papers**: result paper=`2310.11191`, dataset paper=`2104.05767`

**Model**: GPT-4

**Description**

- **Task:**
  - Paragraph-to-document–level text simplification in the medical domain.
  - Input: one or more paragraphs from technical abstracts of clinical evidence reviews.
  - Output: a plain-language, open-ended rewrite that retains key findings.
  - Answer space: unrestricted natural language (no predefined options).
  - Evaluation target: alignment of the generated simplification with the reference plain-language version in content and lexical overlap.
- **Data collection:**
  - Source texts: technical abstracts from evidence syntheses on clinical topics; targets: plain-language summaries authored by domain experts for non-expert audiences.
  - English-only; sourced from a large online repository of medical systematic reviews.
  - Abstracts and corresponding lay summaries were paired via heuristic section matching: results-focused abstract sections were retained; for lay summaries, sections discussing studies/results (or the first paragraph containing terms like "study/trial" for unstructured summaries) were selected. Pairs with large length discrepancies were removed; examples were filtered to fit standard transformer context budgets.
  - Note: lay summaries are not direct simplifications of abstracts (derived from full reviews), yielding partial content overlap rather than sentence-level parallelism.
- **Evaluation:**
  - Metric: ROUGE-LSum (F1, longest common subsequence at the summary level) between model output and the single reference lay summary.
- **Difficulty characteristics:**
  - Domain shift and style gap: targets are stylistically simpler but only marginally easier on readability metrics; success requires controlled paraphrasing and style adaptation beyond sentence/word shortening.
  - Technical jargon and statistical elements (e.g., measurements, confidence intervals) increase lexical/semantic complexity.
  - Non-parallelity: targets may include information absent from abstracts (e.g., currency statements), creating source–target mismatches and risk of hallucination.
  - Long, multi-paragraph inputs necessitate document-level content selection and abstraction.
- **Prompting strategy:**
  - Zero-shot instruction prompting: a system instruction specifies a simplification assistant role; a user instruction asks to simplify the provided text. No examples or chain-of-thought demonstrations are provided.
- **Other settings:**
  - No retrieval augmentation, external tools, or special decoding constraints reported for this record.

**Target metric**: 33.9 (rouge)

**Source Papers**: result paper=`2310.20105`

**Model**: GPT-4

**Description**

- **Task:**
  - Single-label, multi-class text classification of student help requests from introductory programming courses.
  - Input: a natural-language issue description, optionally including a code snippet and an error message; the prompt also provides a numeric indicator of alignment between the message and assignment instructions.
  - Label space: six mutually exclusive categories—three debugging subcategories (error message present; expected outcome specified; both present), plus implementation, understanding, and a nothing/insufficient-content class.
  - Target: assign a single category to each request.
- **Data collection:**
  - Source: requests submitted via a course help assistant using a semi-structured form.
  - Captured contents: issue description, optional code and error message (programming language collected by the form but not used in the prompt).
  - Annotation: two human coders independently labeled requests; disagreements were resolved by adopting the more experienced coder's label; inter-rater agreement reported as substantial.
  - Domain/language/licensing: programming-education context; language unspecified; no licensing details provided.
- **Evaluation:**
  - Metric: F1 score for the specific category (per-class F1) on the test set, comparing model predictions with human reference labels.
- **Subset:**
  - Implementation category only—requests seeking guidance on implementing code for a specific assignment problem (often include code and/or explicit references to instructions).
- **Difficulty characteristics:**
  - Six-way classification with nuanced distinctions based on subtle cues (e.g., presence/absence of code, explicit error messages vs. desired outcomes, references to instructions).
  - Inputs may include code and error traces; no prompt-length constraints were reported to affect evaluation.
- **Prompting strategy:**
  - Zero-shot classification (no in-context examples, no chain-of-thogught).
  - System instruction outlines the full coding scheme and decision criteria (adapted from human-coder guidelines) and requires output of a single category label without explanation.
  - User message includes: the student's issue description, any provided code, any provided error message, and a numeric "issue–instructions equivalence" percentage indicating overlap with assignment instructions (% only used as context).
- **Other settings:**
  - Inference via API, one request at a time.
  - temperature 0.0, top_p 1, max_tokens 10, frequency_penalty 0, presence_penalty 0.

**Target metric**: 85 (F1)

**Source Papers**: result paper=`2406.01238`, dataset paper=`1803.06643`

**Model**: GPT-4

**Description**

- **Task:**
  - Open-domain natural-language question answering over knowledge graphs.
  - Input: a single English question; the system may decompose it into sub-questions and navigate a knowledge graph.
  - Output: one or more target entities/values from the graph; the final prediction is the top candidate.
  - Evaluation target: whether the top predicted answer matches any reference answer.
- **Data collection:**
  - Built by generating complex queries from a simpler QA corpus that pairs questions with graph queries over a large collaborative knowledge graph.
  - Complexity introduced via function composition, conjunctions, superlatives, and comparatives.
  - Machine-generated templates derived from graph predicates were paraphrased by crowd workers into natural language; gold answers obtained by executing graph queries.
  - Domain: broad, open-domain factual knowledge; Language: English; Provenance: curated knowledge graph + crowdsourced paraphrases.
- **Evaluation:**
  - Metric: answer accuracy (Hits@1), i.e., whether the top prediction matches any gold answer string/entity.
- **Difficulty characteristics:**
  - Multi-step reasoning with compositions and set operations (e.g., conjunctions), plus superlative/comparative constraints.
  - Paraphrase variability: human rewrites introduce synonymy, reordering, omissions, and additions.
  - Open-domain coverage and entity-centric reasoning increase ambiguity, requiring precise entity typing and relation selection.
  - Often requires multi-hop traversal over the graph to reach the final answer.
- **Prompting strategy:**
  - Strategic, iterative LLM-driven planning over a knowledge graph (EffiQA). The high-capacity LLM performs: 1) global planning to decompose the question and generate exploration instructions (with simulated answers to guide search); 2) delegated efficient KG exploration via breadth-first expansion with semantic pruning using fine-grained entity typing and similarity to simulated answers; 3) self-reflection to review partial results, revise the plan, and iterate until sufficient evidence is gathered.
- **Other settings:**
  - LLM temperature set to 0 for deterministic outputs.
  - Exploration uses breadth-first search with thresholds to control branching; relations/entities not matching planned types/qualifiers are pruned.
  - A plug-in encoder-based classifier (fine-grained entity typing) supports constrained semantic matching to maintain recall while reducing search space.
  - Stopping criterion: output an answer when results are satisfactory; otherwise replan and iterate (self-reflection loop).

**Target metric**: 69.5 (accuracy)

# F  PROMPTS

## F.1  PROMPT FOR PREDICTION USING SEARCH

---

**Prompt**

Given an experimental record (task, model, and setup), search for supporting evidence and estimate the LLM's performance for the specified setup.

**Inputs**

- Target model: model evaluated on.
- Metric family: metric family used to evaluate the experimental record.
- Redacted description: description of the experimental record (e.g., task, dataset collection, evaluation metric, experimental setup).

**Search Process (REQUIRED)**

- Use the search tool up to four times to gather evidence.
- Keep queries concise; the retrieval module cannot handle complex queries.
- Focus queries on the Target model evaluated on related datasets/tasks (e.g., GPT-4o clinical summarization rouge).
- Submit at most one query per turn; after each query, the tool returns matching paper PDFs.
- Plan what to search before calling the tool.
- If retrieved evidence is irrelevant, do not reference it for the prediction.

**Confidence Assessment**

- After producing a performance prediction, critically review your reasoning and the evidence considered.
- Classify confidence into one of:
  - Almost no chance, Highly unlikely, Chances are slight, Unlikely, Less than even, Better than even, Likely, Very good chance, Highly likely, Almost certain.

**Output Format**

1. Prediction: **Point estimate** on a 0–100 scale as \boxed{}.
2. Rationale: brief, ordered reasons mapping features → expected performance shifts (e.g., 5 options (20% baseline) + few-shot CoT (+) + domain shift (–) + exact match (–) → net expectation . . . ).
3. Confidence Assessment: one class name wrapped in \confidence{}.
4. Retrieval Evidence: list search queries, retrieved papers, and concise summaries of relevant findings from those papers.

---

## F.2 PROMPT FOR PREDICTION WITHOUT USING SEARCH

**Prompt**

Given an experimental record (task, model, and setup), estimate the LLM's performance for the specified setup.

**Inputs**

- Target model: model evaluated on.
- Metric family: metric family used to evaluate the experimental record.
- Redacted description: description of the experimental record (e.g., task, dataset collection, evaluation metric, experimental setup).

**Confidence Assessment**

- After producing a performance prediction, critically review your reasoning and the evidence considered.
- Classify confidence into one of:
  - Almost no chance, Highly unlikely, Chances are slight, Unlikely, Less than even, Better than even, Likely, Very good chance, Highly likely, Almost certain.

**Output Format**

1. Prediction: **Point estimate** on a 0–100 scale as \boxed{}.

2. Rationale: brief, ordered reasons mapping features $\rightarrow$ expected performance shifts (e.g., 5 options (20% baseline) + few-shot CoT (+) + domain shift (–) + exact match (–) $\rightarrow$ net expectation . . . ).

3. Confidence Assessment: one class name wrapped in \confidence{}.

4. Retrieval Evidence: list search queries, retrieved papers, and concise summaries of relevant findings from those papers.

## G    COMPLIANCE WITH ARXIV ACCESS POLICIES

We accessed arXiv content in accordance with arXiv's published guidance for automated use, including the `robots.txt`[2] and the "Robots Beware" help page.[3] Our goal was to minimize load on arXiv's infrastructure while ensuring traceable, policy-compliant retrieval.

**Scope of access.**    Agents fetched only URLs permitted by `robots.txt` and avoided disallowed paths. We did not perform indiscriminate crawling or attempt corpus-scale downloads via the human-facing site.

**Preferred interfaces.**    Wherever possible, we used arXiv's programmatic interfaces (e.g., API/OAI-PMH/RSS) instead of scraping HTML pages, consistent with the help page's guidance to favor machine interfaces for automated access.[2]

**Robots rules and rate limiting.**    Our clients honored `robots.txt` directives (allow/disallow and crawl-delay) and enforced conservative global throttling with backoff on errors. We limited parallelism and maintained a single logical connection per agent run.

**Identification and advance notice.**    All requests included a descriptive `User-Agent` string and a monitored contact email, as recommended.[2] Prior to large-scale experiments, we emailed the arXiv team responsible for automated access to describe our usage pattern and confirm that no special coordination was required.

**Low-impact retrieval policy.**    To further reduce load, each retrieval-capable agent capped (i) the number of queries it could issue and (ii) the number of documents returned per query. We cached responses, deduplicated URLs, and normalized links to canonical arXiv endpoints to avoid redundant traffic.

**Monitoring and fail-safe behavior.**    We continuously monitored HTTP status codes and immediately slowed or halted traffic upon receiving `429/403/5xx` responses. Our clients also rechecked `robots.txt` at regular intervals to respect policy changes; any change that tightened restrictions triggered an automatic stop.

**Summary.**    In aggregate, our practices align with arXiv's automated access rules: we honored `robots.txt`, preferred official machine interfaces, rate-limited conservatively, identified ourselves, notified arXiv in advance, and implemented additional safeguards to minimize load.

## H    LLM USAGE DISCLOSURE

We disclose all uses of large language models (LLMs) in the preparation of this work.

**Scope of use**    LLMs were used *only* for (i) **data curation utilities** and (ii) **baseline models** in our experiments, and (iii) **light writing assistance**. LLMs were *not* used for research ideation, experimental design, claim formation, or result interpretation.

**Data curation**    LLMs assisted with mechanical curation tasks such as dataset source annotation, extraction, and rewriting. All curation outputs were reviewed by the authors; any labels or annotations used for evaluation were not generated solely by LLMs.

**Baselines**    As reported in the main text (e.g., table 3), we evaluate several off-the-shelf LLMs strictly as *baselines*. These models were used via their standard inference APIs with fixed prompts and hyperparameters (temperature, top-$p$, max tokens) specified in the paper/appendix. No additional finetuning of these baseline LLMs was performed.

---

[2]https://arxiv.org/robots.txt
[3]https://info.arxiv.org/help/robots.html

**Writing assistance**   LLMs were used for *copy-editing* only: grammar checks, minor rephrasing of awkward sentences, and generating LaTeX boilerplate (e.g., table skeletons). All scientific content, analysis, and conclusions are authored by the human authors; every LLM-suggested edit was manually reviewed and, when necessary, revised.

**Non-use**   LLMs were *not* used for ideation or synthesis of contributions. They did not select experiments, decide hypotheses, or write technical sections beyond copy-editing as described above.

**Privacy and safety**   No private or personally identifiable information was provided to LLMs. Inputs were limited to public research materials and our own synthetic/derived artifacts. API usage complied with the respective providers' terms of service.

**Author responsibility**   The authors take full responsibility for the paper's content. LLM assistance was limited to the bounded uses above and did not alter the scientific claims or their justification.

