# OpenReview forum: "Look Before you Leap: Estimating LLM Benchmark Scores from Descriptions"
_ICLR.cc/2026/Conference — Submitted to ICLR 2026_

### Official Review · Reviewer_EXyJ · 2025-10-27

**Soundness:** 2
**Presentation:** 2
**Contribution:** 3
**Rating:** 4
**Confidence:** 3

**Summary:**

This paper explores the possibiltiy of predicting an LLM's performance on a new benchmark, solely based on a textual description of the evaluation setup. The authors introduce a corresponding corpus, datamined from arxiv. On that corpus, zero-shot GPT-5 clearly outperforms a constant baseline, while Qwen models generally perform close to the constant baseline in terms of MAE. Allowing models to search arxiv papers (excluding the paper containing the prediction target) has inconsistent benefits, with minor improvements for GPT-5, but no clear trend for Qwen models.

**Strengths:**

- The paper considers an interesting setting, and the results look promising
- The authors seem to have put care into making their crawling responsible and complying with the arxiv access policies

**Weaknesses:**

- Despite some efforts to combat task leakage, this remains a major concern: If I understand correctly, the corpus includes papers from 2023 to 2024, while GPT5's knowledge cutoff is in June 2024, suggesting that it might have "seen" most of the papers used in the main benchmark during training.
- Reproducibility is generally low. For example, the specific parameters for GPT-5 queries do not seem to be documented. Similarly, it is not completely clear whether Qwen3 thinking was ran with greedy decoding (which is explicitly discouraged in the HF docs and might explain performance issues), or the default parameters.
- Given the brittleness of LLM benchmark scores, the prediction task seems to be potentially underspecified. For example, the specific way in which classification scores are extracted from LLMs is known to cause large differences in benchmarks scores (https://huggingface.co/blog/open-llm-leaderboard-mmlu), but does not seem to be included in the example dataset items. Similarly, in many cases, it seems plausible that up/downweighing "harder" items within the boundaries of the description could strongly affect benchmark performance.
   - While it is perfectly fine for a benchmark to be underspecified in this sense, the relatively strong performance despite these issues exacerbates concerns about spurious features and contamination.
- Additional ablations and experiments would be useful to evaluate the paper's cliams.
   - To isolate the impact of reasoning, GPT-5 and Qwen3 could be evaluated with and without reasoning, rather than comparing older, non-reasoning models with newer models.
    - The streaming experiment could be rerun without search enabled (on the same papers)
    - Evidence against contamination could be provided by evaluating models' ability to predict the dataset name from the provided description

**Questions:**

- "Because source fidelity is critical, any remaining errors are corrected via targeted human edits." Was this done for *all* papers, or a subset?
- "we focused on curating diverse and high- quality examples using the GPT-5 (OpenAI, 2025) API ($200) " Does this simply refer to the process detailed in the paper, or was there additional "curation"?
- For the streaming setup, what steps were taken to ensure papers were processed before being indexed by GPT's web search?
- Regarding the text-encoder pipeline, what was the value of k for the cross-validation, and how were the baselines tuned? Also, how was E5-Mistral-7B chosen as the encoder?
- Do I understand correctly, that most benchmarks in the dataset only have a single model evaluated on them?
- Is there a reason to lower-bound MAE at 10 in Figure 3?

---

> ### Author Response · Authors · 2025-11-21
> **Response to Reviewer EXyJ [1/2]**
>
> Thank you for the detailed and constructive comments. We address your concerns in below.
>
> > The corpus includes papers from 2023 to 2024, while GPT5's knowledge cutoff is in June 2024, suggesting that it might have "seen" most of the papers during training
>
> Yes, we also viewed this as a critical issue, so we reconstructed PRECOG using a arXiv 2025 snapshot (January-September) and re-ran all experiments on a leakage-controlled PRECOG 2025 (with GPT-5 and Qwen3 knowledge cutoff in 2024).
>
> The updated dataset on arXiv 2025 (Jan-Sep) is larger and more diverse:
>
> | **Statistic** | **Value** |
> | --- | --- |
> | Total records (length of dataset) | 1,523 |
> | Records with TSP = DSP | 549 |
> | Records with TSP ≠ DSP | 974 |
> | Unique table-source arXiv IDs | 579 |
> | Unique dataset-source arXiv IDs | 616 |
> | Unique union of table- and dataset-sources | 956 |
>
> Main results on this leakage-controlled version:
>
> | **Method** | **MAE ↓** | **Pearson r ↑** |
> | --- | --- | --- |
> | Estimation-set mean | 23.0 | – |
> | E5-Mistral + kNN | 21.2 | 0.22 |
> | E5-Mistral + XGBoost | 21.4 | 0.20 |
> | GPT-5, medium reasoning | **14.6** | 0.52 |
> | GPT-5, minimal reasoning | 15.5 | 0.62 |
> | Qwen3-32B, with reasoning | **19.6** | 0.41 |
> | Qwen3-32B, no reasoning | 22.5 | 0.37 |
>
> Even under this stricter, leakage-controlled setup, GPT-5 and Qwen3 still substantially outperform both the mean baseline  in terms of MAE and correlation, indicating that short descriptions plus literature context retain non-trivial predictive signal. Reasoning-style prompting also consistently helps (medium vs minimal for GPT-5, reasoning vs no reasoning for Qwen3), suggesting the models are actively using the textual cues rather than memorizing. Sections 4, 5, and 6 have been reorganized to include the 2025 results, which are also reflected in Tables 2 and 3.
>
> > Reproducibility is generally low.
>
> We reran the experiments with the specified prompting setup and random seed, and have updated the paper accordingly. For GPT-5, we used the default decoding hyperparameters, and for Qwen3 we followed the decoding settings recommended by the original model authors (not greedy decoding). We have clarified this in L279 and L289. All code and data required to reproduce our experiments will be released.
>
> > Given the brittleness of LLM benchmark scores, the prediction task seems to be potentially underspecified.
>
> We agree that benchmark scores are brittle and that our task is inherently noisy. Our goal is *not* to replace standard evaluations or to predict exact final scores. Instead, the system is designed as a **pilot-stage tool** that provides a coarse but informative signal about where to invest time, annotation effort, and GPU compute before running expensive experiments. In that context, an MAE in the mid-teens with non-trivial correlation is still useful: it allows practitioners to distinguish clearly underperforming regimes from plausibly viable ones, rather than to finely rank nearly identical models.
>
> For example, a team can use our framework to see whether a support-ticket summarization model is likely in the “about 50–60” range or closer to “about 80.” Even with ~14 MAE, the gap between “mid-50s” and “mid-80s” is large enough to guide proceed-or-not and resource decisions. We have added a discussion of this limitation in L511 and included a concrete case study of our work in Appendix D.
>
> > GPT-5 and Qwen3 could be evaluated with and without reasoning.
>
> We have added this ablation study in Table 3. For GPT-5, we control reasoning through the API’s *reasoning_effort* parameter and compare minimal vs medium. There is currently no way to turn reasoning fully off, so “minimal” is the closest setting to a no-reasoning condition, which is why we used it. For Qwen, we switch between “thinking” and “no thinking” by adding or removing a special token in the tokenizer, which turns its reasoning mode on or off.
>
> The results are:
>
> | **Method** | **MAE ↓** | **Pearson r ↑** |
> | --- | --- | --- |
> | GPT-5, medium reasoning | 14.6 | 0.52 |
> | GPT-5, minimal reasoning | 15.5 | 0.62 |
> | Qwen3-32B, with reasoning | 19.6 | 0.41 |
> | Qwen3-32B, no reasoning | 22.5 | 0.37 |
>
> Reasoning-style prompting consistently helps for both models: it reduces error for Qwen3 and improves robustness and correlation for GPT-5, indicating that the models are using the textual descriptions in a non-trivial way rather than simply memorizing.

---

> ### Author Response · Authors · 2025-11-21
> **Response to Reviewer EXyJ [2/2]**
>
> > The streaming experiment could be rerun without search enabled.
>
> We reran the streaming experiment without search and added these results to the paper (GPT-5 only) in Table 7:
>
> | **Method** | **MAE ↓** | **Pearson r ↑** |
> | --- | --- | --- |
> | Test-set mean baseline | 25.5 | – |
> | GPT-5 with search | 14.0 | 0.78 |
> | GPT-5 without search | 15.09 | 0.74 |
>
> Search improves performance, but even without search the predictor retains strong correlation and substantially lower MAE than the baseline.
>
> > Evidence against contamination could be provided by evaluating models' ability to predict the dataset name.
>
> We ran the suggested analysis by asking GPT-5 to predict the dataset name from our descriptions and manually validating the outputs, for two subsets:
>
> | **Subset** | **Correct / Total** |
> | --- | --- |
> | Benchmark records | 6.6% |
> | Experimental records | 38% |
>
> As expected, name recovery is very low for benchmark-introducing papers (which mostly present new datasets) and higher for experimental records that reuse well-known benchmarks. In the remaining 6% of benchmark records (two cases), the table source papers adapted existing benchmarks, but the LLM mistakenly treated those papers as if they introduced the benchmarks. This indicates that the model can sometimes recognize famous benchmarks when they are reused, but rarely recovers the names of truly new datasets.
>
> Crucially, when we split our main results by these two regimes, we do not see a large difference in prediction quality:
>
> | **Model** | **Regime** | **n** | **MAE ↓** | **Pearson r ↑** |
> | --- | --- | --- | --- | --- |
> | GPT-5 | Experimental setup | 973 | 14.72 | 0.69 |
> | GPT-5 | Benchmark setup | 549 | 15.12 | 0.67 |
> | Qwen3 | Experimental setup | 974 | 19.11 | 0.42 |
> | Qwen3 | Benchmark setup | 549 | 20.71 | 0.40 |
>
> Performance is very similar between “experimental setup” entries (using existing benchmarks) and “benchmark setup” entries (introducing new benchmarks). This suggests that while the model can sometimes recognize well-known benchmark names, its predictive behavior is not restricted to canonical, long-established benchmarks and also extends to newly introduced, previously unseen datasets. We have added this subset analysis in Table 4.d.
>
> > When constructing the dataset, were errors fixed with targeted human edits for all papers, or only for a subset?
>
> Yes. We applied this procedure to all cases where the table’s source paper differed from the dataset’s source paper. Although we initially used the Qwen3-32B model for this task, the error rate dropped to under 2% after switching to GPT-5-mini when constructing PRECOG 2025. Remaining errors were largely eliminated during the description-extraction step, where the model was prompted to output “fail” whenever the source was incorrect. As a result, the pipeline is now fully automated and don’t require targeted human edits.
>
> > We focused on curating diverse and high- quality examples using the GPT-5" … Does this simply refer to the process detailed in the paper, or was there additional "curation"?
>
> For curation, we applied targeted human edits to the source data and performed manual dataset name normalization, using GPT-5 for extracting description. During PRECOG 2025 curation however, we found that GPT-5 mini achieved competitive curation performance, so we switched to GPT-5 mini which is strong and cost-efficient. We also used dataset source papers to perform hierarchical sampling of instances to cover diverse datasets, which removed the need for dataset name normalization. We clarified this in L814.
>
> > For the streaming setup, what steps were taken to ensure papers were processed before being indexed by GPT's web search?
>
> First, we used only newly released arXiv IDs when generating descriptions and predictions, and ran the pipeline within about four hours of release to avoid leakage. In addition, because GPT’s web search can return documents it has just browsed, we used arXiv IDs and content checks to detect when the model had accessed a target paper and dropped those cases, filtering out about 35% of instances. We clarified this is 487.
>
> > Regarding the text-encoder pipeline, what was the value of k , and how were the baselines tuned? Also, how was E5-Mistral-7B chosen as the encoder?
>
> We set K = 10 and used E5-Mistral-7B, which is a strong state-of-the-art dense embedding encoder baseline. Hyperparameters were tuned on a held-out validation split from the 2023–2024 version of PRECOG. We added this description in L278.
>
> > Do I understand correctly, that most benchmarks in the dataset only have a single model evaluated on them?
>
> Yes, mostly. We focused on two models, GPT-4 and GPT-4o, so we could study dataset and prompt effects. Other models have too little data, while GPT-4 and GPT-4o are the ones with many published records. While some instances of the same benchmark include results for both models, in most cases each benchmark is evaluated with only a single model.

---

> > ### Comment · Reviewer_EXyJ · 2025-11-21
> >
> > Thank you, this seems to address most of my concerns. I have tentatively raised my score.
> >
> > Some remaining questions:
> >  - Is E5-Mistral-7B really state of the art? I am no expert in text-embedding models, but it seems quite small and more than one year old.
> >  - Do you expect this to work well when targeting newer models, or do you think that LLM's prediction performance on these tasks mostly stems from the abundance of GPT-4 evaluations published in the literature?

---

> > > ### Author Response · Authors · 2025-11-21
> > > **Response to Reviewer Exyj**
> > >
> > > We sincerely thank you for constructive feedbacks and for raising the scores generously.
> > >
> > > We address your questions in below:
> > >
> > > > Is E5-Mistral-7B really state of the art? I am no expert in text-embedding models, but it seems quite small and more than one year old.
> > >
> > > Thank you for pointing this out. We agree that calling E5-Mistral-7B “state of the art” is too strong given the rapid progress in text-embedding models. In the revision, we now describe it more carefully as a **strong open-weight 7B embedding model that was among the top-performing models on MTEB at the time we started our experiments**, and we have removed the “state-of-the-art” wording from the main text.
> > >
> > > To address your concern more concretely, we also ran additional experiments on the **PRECOG-2025** dataset using more recent embedding backbones that are competitive or state-of-the-art on MTEB leaderboard, including **Qwen3-8B-Embed** and **KaLM-Embedding-Gemma3-12B-2511**, with both KNN and XGBoost regressors. The results are summarized below:
> > >
> > > | **Model** | **MAE** | **Pearson correlation** |
> > > | --- | --- | --- |
> > > | E5-Mistral-7B + KNN | 21.2 | 0.22 |
> > > | E5-Mistral-7B + XGBoost | 21.4 | 0.20 |
> > > | Qwen3-8B-Embed + KNN | 20.8 | 0.27 |
> > > | Qwen3-8B-Embed + XGBoost | 21.6 | 0.21 |
> > > | KaLM-Embedding-Gemma3-12B-2511 + KNN | 20.8 | 0.26 |
> > > | KaLM-Embedding-Gemma3-12B-2511 + XGBoost | 21.5 | 0.21 |
> > >
> > > As the table shows, **newer and stronger embedding models somewhat modest improvements, but the gains are marginal**. We added this results to the Appendix C (L936).
> > >
> > > > Do you expect this to work well when targeting newer models, or do you think that LLM’s prediction performance on these tasks mostly stems from the abundance of GPT-4 evaluations published in the literature?
> > >
> > > We appreciate this important question and have added clarification around this point in the revised version.
> > >
> > > First, our analysis in Table 4.c indeed shows that the **model (GPT-4) with a larger number of published experiments tend to be somewhat easier to predict**: performance improves as more historical evaluations are available. Intuitively, this is expected—our method learns from patterns in past experiments, so having more observations per model provides more signal.
> > >
> > > That said, we agree that **generalizing to truly “new” models with very few or no prior evaluations is an important limitation and a natural direction for future work**. We now emphasize this point explicitly in the discussion section (L536): making the approach robust for newer models that are not well covered before the knowledge cutoff (e.g., via better priors, active collection of seed evaluations, or hybrid text–metadata representations) is a key open problem that we are excited to explore in follow-up work.

---

### Official Review · Reviewer_3Gxd · 2025-10-31

**Soundness:** 1
**Presentation:** 3
**Contribution:** 2
**Rating:** 2
**Confidence:** 2

**Summary:**

This paper studies text-only performance forecasting: the task of estimating a model’s benchmark score solely from a natural-language description of the task and configuration, without running the experiment. The authors introduce PRECOG, a dataset of redacted description–performance pairs built from arXiv papers, and evaluate several LLMs (GPT-5, Qwen3-32B, etc.) as regressors.
They show that only GPT-5 achieves meaningful accuracy, outperforming humans and classical baselines, while open-source models remain close to simple predictors (test-set average, text-encoder based regressors).

**Strengths:**

- The paper presents an interesting framing of model evaluation as a forecasting problem.
- The PRECOG dataset appears well documented and carefully constructed, with redaction, leakage control, and validation steps.
- The authors develop their own retrieval-based tool that allows models to access relevant literature before making predictions, which both enriches the setup and helps ensure a fairer comparison across models.

**Weaknesses:**

- While the framing of “evaluation as forecasting” is conceptually interesting, the concrete benefits of text-only performance prediction are not fully demonstrated.  The reported errors (e.g, 14.0 MAE with GPT-5 + Search on a $\[ 0, 100 \]$ scale) suggest forecasts may be too coarse to guide real experimental decisions.  The work would be stronger with a case study illustrating how such forecasts could meaningfully inform benchmark design, compute allocation, or early-stage experiment planning.
- Current non LLM baselines (mean predictor, simple regressors) are rather weak, yet only GPT-5 manages to outperform them. This makes it hard to assess the usefulness of the benchmark itself, since improvements may only reflect pretrained knowledge rather than genuine reasoning ability.
- The human baseline is interesting but too limited to show practical utility. The sample is small, lacks calibration, and participants do not have the same background information as GPT-5. This makes the gap hard to interpret and insufficient to support claims of useful forecasting ability.
- The analysis of retrieval and reasoning patterns (e.g., number of queries, token diversity) remains mostly descriptive. More thorough inspection could reveal whether retrieval genuinely contributes to improved forecasts, what kinds of evidence are most useful, and how reasoning strategies differ between models

**Questions:**

- How do you envision these forecasts becoming practically useful for researchers? For example, what level of accuracy or reliability would make them meaningful for experiment planning or benchmark design?
- You mention several leakage-control measures in constructing PRECOG, but it remains challenging to determine whether an LLMs’ forecasting accuracy reflects genuine reasoning about task descriptions or residual background knowledge from pretraining. Could you elaborate on how your controls address this issue and whether there is room for stronger evaluation of reasoning beyond memorized benchmark patterns?
- Could you elaborate on how retrieval influences forecasting performance and what evidence supports its contribution to improved results?

---

> ### Author Response · Authors · 2025-11-21
> **Response to Reviewer 3Gxd [1/3]**
>
> Thank you for the helpful and constructive comments. We address each of your points below.
>
> > The concrete benefits of text-only performance prediction are not fully demonstrated… How do you envision these forecasts becoming practically useful for researchers?
>
> Thank you for raising this point. Our goal is not to replace standard evaluations, but to provide a coarse, text-only forecasting signal that helps decide where to spend time, annotation budget, and GPU compute *before* running expensive sweeps. In that setting, an MAE in the mid-teens with non-trivial correlation is still useful: it lets researchers distinguish clearly underperforming regimes from plausibly viable ones, rather than finely rank nearly identical models. Concretely, the intended use is to **screen and prioritize**, not to predict exact final scores.
>
> We present a brief case study to illustrate how our method can be useful in practice. The puzzle-show benchmark [1] offers a concrete quantitative example in a setting where the original authors already suspected that off-air weekly challenges would be harder than on-air live puzzles. In this benchmark, GPT-4o reaches about 65% accuracy on the easier on-air subset and about 6% on the harder off-air subset. Using only natural-language task descriptions (for example, “live, quickly solvable on-air puzzles answered in seconds by callers” versus “harder, week-long off-air challenges that typical listeners may think about for days”), our predictor outputs 68% (on-air) and 14% (off-air) for GPT-4o, with absolute errors of 3 and 8 percentage points. The predictor correctly predicts a large performance drop on the off-air version before the actual off-air score is observed. In a hypothetical design scenario where neither subset has been built, a benchmark designer could write descriptions for an “easy/live” variant and a “hard/take-home” variant, query our system once, and use the predicted gap (68 vs. 14) as ex-ante evidence that the harder off-air set is both feasible and not saturated, which would help prioritize data collection and curation for that variant before running costly pilot experiments on both. We added a more detailed discussion of this case study in Appendix D.
>
> [1] PhD Knowledge Not Required: A Verbal Reasoning Challenge for Large Language Models, Wu et al., 2025
>
> | Subset | Gold accuracy (%) | Predicted accuracy (%) | Absolute error |
> | --- | --- | --- | --- |
> | On-air | 65 | 68 | 3 |
> | Off-air | 6 | 14 | 8 |
> | Overall | – | – | 5.5 (mean absolute error) |
>
> > Limitations of the human baseline.
>
> Our goal with the human study is to show that this forecasting task is genuinely hard even for informed users. The participants were ML/NLP graduate students, so their performance is a reasonable proxy for how current expert users of LLMs might predict benchmark outcomes. With more practice they might improve somewhat, but people are known to struggle with judging probabilities and making decisions under uncertainty (as shown in [2]), and our results are in line with that: even under these constraints, humans find it hard to produce consistent coarse forecasts, while GPT-5 without search achieves clearly lower error and higher correlation on the same inputs. We have added detailed discussion in L411.
>
> [2] Subjective probability: A judgment of representativeness, Kahneman and Tversky, Cognitive psychology 1972

---

> ### Author Response · Authors · 2025-11-21
> **Response to Reviewer 3Gxd [2/3]**
>
> > Improvements may only reflect pre-trained knowledge rather than genuine reasoning ability… Could you elaborate on how your controls address this issue and whether there is room for stronger evaluation of reasoning beyond memorized benchmark patterns?
>
> Thank you for raising this concern. We agree that, in principle, the gains could come from memorized benchmark results rather than genuine use of the text descriptions. To address this, we ran the streaming prediction analysis in Section 8 to go beyond memorized benchmark patterns. We also ran additional experiments to test this, which we show below.
>
> First, we re-ran all experiments on a **leakage-controlled PRECOG 2025 snapshot** (with GPT-5 and Qwen3 knowledge cutoff set to 2024), where newly added records post-date the models’ training horizon. The updated dataset on arXiv 2025 (Jan-Sep) is larger and more diverse.
>
> | **Statistic** | **Value** |
> | --- | --- |
> | Total records (length of dataset) | 1,523 |
> | Records with TSP = DSP | 549 |
> | Records with TSP ≠ DSP | 974 |
> | Unique table-source arXiv IDs | 579 |
> | Unique dataset-source arXiv IDs | 616 |
> | Unique union of table- and dataset-sources | 956 |
>
> | **Method** | **MAE ↓** | **Pearson r ↑** |
> | --- | --- | --- |
> | Estimation-set mean | 23.0 | – |
> | E5-Mistral + kNN | 21.2 | 0.22 |
> | E5-Mistral + XGBoost | 21.4 | 0.20 |
> | GPT-5, medium reasoning | 14.6 | 0.52 |
> | GPT-5, minimal reasoning | 15.5 | 0.62 |
> | Qwen3-32B, with reasoning | 19.6 | 0.41 |
> | Qwen3-32B, no reasoning | 22.5 | 0.37 |
>
> Even in this stricter setting, GPT-5 and Qwen3 substantially outperform both the mean baseline and embedding-based regressors in MAE and correlation, suggesting that they are extracting useful signal from the descriptions and literature context, not just recalling scores. Moreover, **reasoning-style prompting consistently helps** (medium vs. minimal for GPT-5, reasoning vs. no reasoning for Qwen3), which is hard to explain purely by memorization of benchmark numbers and instead points to the model using the textual cues to organize its prediction.
>
> Second, we examined whether performance is limited to “seen” or canonical benchmarks. We split entries into **experimental records** (papers reusing existing benchmarks) and **benchmark records** (papers introducing the dataset):
>
> | **Model** | **Regime** | **n** | **MAE ↓** | **Pearson r ↑** |
> | --- | --- | --- | --- | --- |
> | GPT-5 | Experimental records | 973 | 14.7 | 0.69 |
> | GPT-5 | Benchmark records | 549 | 15.1 | 0.67 |
> | Qwen3 | Experimental records | 974 | 19.1 | 0.42 |
> | Qwen3 | Benchmark records | 549 | 20.7 | 0.40 |
>
> Performance is very similar across these two regimes. In addition, on **newly curated, truly unseen datasets** from the 2025 arXiv snapshot, GPT-5 and Qwen3 still achieve a Pearson correlation of 0.66. This indicates that the system is not only recalling known benchmarks, but also providing meaningfully aligned predictions for datasets and setups that are unlikely to have been seen during pre-training.
>
> We do not claim that our method exposes “pure reasoning” or that 14–15% MAE is perfect. Our claim is more modest: even under leakage-controlled and unseen settings, text-only forecasts remain competitive and well correlated with true scores, which is sufficient for our intended use of **coarse, early-stage screening and prioritization** rather than replacing full evaluations. We have added the 2025 snapshot results to Tables 2 and 3, and the experimental records versus benchmark records to Table 4.d. Additionally, we have included a discussion of limitations in L511 and an interesting use case in Appendix D.

---

> ### Author Response · Authors · 2025-11-21
> **Response to Reviewer 3Gxd [3/3]**
>
> > The analysis of retrieval and reasoning patterns remains descriptive… Could you elaborate on how retrieval influences forecasting performance and what evidence supports its contribution to improved results?
>
> Thank you for this question. we agree that the role of retrieval needs to be made more explicit.
>
> In our revision we distinguish two settings and report both quantitative and qualitative evidence:
>
> **Offline PRECOG setting with a custom retriever.**
>
> We use a sparse retriever that is forced not to return the target paper, and we compare GPT-5 and Qwen3 with and without retrieval. The gains are small and sometimes negative, especially for Qwen3. When we asked an LLM to rate the retrieved documents on a 5-point relevance scale, most were only “moderately relevant.” In practice, the retriever often surfaces high-level related work, not papers that add much beyond the model’s pretraining and our structured description. This explains why retrieval has little effect and can even hurt Qwen3 in some cases. Overall, retrieval is not the main source of forecasting gains in PRECOG; most of the signal comes from reasoning over the curated descriptions plus the model’s own knowledge.
>
> **Streaming setting with web search.**
>
> In contrast, when we allow GPT-5 to use OpenAI’s web search in the streaming prediction experiment, retrieval clearly helps. The ablation is:
>
> | **Method** | **MAE ↓** | **Pearson r ↑** |
> | --- | --- | --- |
> | Test-set mean baseline | 25.5 | – |
> | GPT-5 with search | 14.0 | 0.78 |
> | GPT-5 without search | 15.09 | 0.74 |
>
> The system with search both reduces error and increases correlation, especially on very recent arXiv entries that the model has not seen during pre-training. This suggests that *strong* retrieval, when it can find genuinely informative context (e.g., related benchmarks or prior results), can meaningfully sharpen forecasts. However, this web search is also a sophisticated, general tool that may surface the target paper itself, which raises leakage concerns that we sought to avoid in the main PRECOG setup.
>
> Putting these together, our current evidence supports the following picture, which we now emphasize in Section 7 (L454) and Section 8:
>
> - **Reasoning over structured descriptions + parametric knowledge is doing most of the work** in the main PRECOG 2025 experiments; the leakage-safe, sparse retriever we built adds only modest and sometimes inconsistent gains.
> - **Retrieval can substantially improve forecasting when it is strong and unconstrained**, as in the streaming experiment with web search, especially for very new or out-of-distribution tasks — but this comes with a real risk of contaminating the prediction by directly accessing the target paper.

---

> > ### Comment · Reviewer_3Gxd · 2025-11-27
> >
> > Thank you for the detailed responses and additional experiments. Many of my earlier questions, particularly those about retrieval and memorization, are now clearer, and the new results and explanations help address some of the concerns I raised.
> >
> > However, my main reservation still remains around the demonstration of concrete practical benefit. The case study feels somewhat post-hoc, and it is not yet clear how the method would guide decision-making or reduce effort in a realistic workflow. The feasibility claim is also a little difficult to interpret: the true accuracy on the hard subset is 6%, whereas the absolute prediction error is 8 percentage points, so I am not sure this strongly supports the claim that the model identifies feasibility.
> >
> > In parallel, I am still not fully convinced about the strength and coverage of the baselines. It remains unclear whether simpler, more task-aligned baselines might achieve similar performance in these practical settings.
> >
> > Overall, while several of my earlier issues were addressed, this question of demonstrating practical utility remains the key concern for me. I think the work would benefit from a more systematic or prospective evaluation of how the predictions actually influence decisions, prioritization, or resource allocation. Strengthening this aspect would significantly improve the paper.

---

> > > ### Author Response · Authors · 2025-11-28
> > > **Response to Reviewer 3Gxd**
> > >
> > > We thank the reviewer for the constructive feedback, which prompted us to add a more concrete, forward-looking example of practical utility. In the revised version (Appendix D.2), we introduce a prospective case study based on a multitask multiple-choice benchmark in the style of MMLU-Pro, focusing on one design configuration: the number of answer options per question. We consider a designer who has not yet built the dataset and is choosing between 5, 10, 15, 20, or 25 options. Using textual descriptions of these five variants, our predictor (GPT-5) estimates GPT-4o accuracies as in the table.
> > >
> > > | # answer choices | Predicted accuracy (%) |
> > > | --- | --- |
> > > | 5 | 86 |
> > > | 10 | 70 |
> > > | 15 | 65 |
> > > | 20 | 62 |
> > > | 25 | 60 |
> > >
> > >
> > > This example is meant to clarify how the method can guide decisions in practice. Before collecting any data, the designer can use a single batch of predictions to rule out clearly saturated settings (5 options), avoid overly difficult ones (20–25 options), and focus annotation and compute on intermediate configurations (around 10–15 options) where performance is expected to fall in a useful range.

---

### Official Review · Reviewer_zA62 · 2025-11-01

**Soundness:** 3
**Presentation:** 3
**Contribution:** 3
**Rating:** 6
**Confidence:** 3

**Summary:**

This paper introduces a framework to predict a model’s benchmark performance without explicitly evaluating the model or viewing the model's ability in source papers. They are able to accurately forecast model abilities. They also validate their findings on newly released datasets and experiments.

**Strengths:**

I like Table 1 with clear differentiation from prior work. I also think the systems seem quite useful for accelerating the development and research of LLMs. I was also impressed by the streaming prediction and human baseline.

**Weaknesses:**

Redundant related work section. I’d also prefer this to be near the beginning if kept.

I’d like to see more cost/resource analysis presented. What are the resource requirements in terms of time and money for all the different baselines? How does this compare to running the actual model on the baseline?

I’d like more discussion of the weaknesses and limitations. Do I never have to run benchmarks again? Where and when should I use this system? On all models for all benchmarks? There is some discussion in the "further analysis across data subsets" part; this should be clarified to see the big picture

**Questions:**

The highest gains are seen with GPT-5 + search, with reasoning. This is a somewhat expensive model to run. How does this compare with actually running the benchmark for standard benchmarks?

Is there any heterogeneity predicting the performance across different benchmarks and different models (ie, open vs closed source models, reasoning vs knowledge benchmarks, recent vs old models, small vs large, etc)?

---

> ### Author Response · Authors · 2025-11-21
> **Response to Reviewer zA62 [1/2]**
>
> Thank you for your constructive feedback. We address each of your points below.
>
> > Redundant related work section.
>
> We merged the original Table 1 (comparison of related work) into the Related Work section and moved this discussion to the beginning of the paper. We also updated both the table and the text to include more recent studies (L140), improving clarity and coverage while avoiding redundancy.
>
> > I’d like to see more cost/resource analysis presented.
>
> The average cost of performance prediction (that is, making a prediction from the description) with GPT-5 (medium reasoning, no search) is about $0.01 per query when using the Batch API. For Qwen models, we run a single A40 GPU inference with a maximum generation length of 32,768 tokens. In terms of latency, the Batch API can process many queries in parallel; on average, a single GPT-5 prediction takes about 132 seconds, while a single Qwen prediction takes about 48 seconds.
>
> Although it is hard to directly compare this to the cost of full evaluations, description-based performance prediction is clearly much more cost-efficient, even if it is less fine-grained and not perfectly accurate. Our goal is to use these predictions in the pilot phase (before finalizing datasets and running full experiments) to design a more cost-effective workflow. We have added these resource-related details and discussion to Section 9 (L521).
>
> > I’d like more discussion of the weaknesses and limitations.
>
> Thank you for raising this. We agree that the big picture and limitations should be clearer. Our goal is not to replace benchmarks or suggest that one never has to run them again. **The system is meant as a pilot-stage tool for rough screening and prioritization**: estimating the difficulty of new datasets or experimental setups, narrowing a large design space to a smaller set of promising configurations, and supporting early go/no-go decisions for new tasks or deployments. It is not intended for finely ranking closely matched models, replacing full benchmark evaluations, or being applied blindly to all models and benchmarks.
>
> We have made this positioning explicit in the paper (L511). In particular, we have emphasized that an MAE in the mid-teens with moderate–high correlations is too coarse to decide whether one model is 73.2% vs. 74.0%, but sufficiently informative to distinguish clearly weak regimes (e.g., 40–50) from potentially deployment-ready regimes (e.g., around 80-90). We have also clarified technical limitations: predictions depend heavily on the quality and specificity of the textual descriptions; the method is only appropriate for coarse, early-stage decisions rather than leaderboard-level comparisons; and, despite our leakage controls (e.g., PRECOG 2025), the predictor can still reflect biases in its training distribution. We additionally include a useful case study of our work in Appendix D.
>
> > How does GPT5+Search compare with actually running the benchmark for standard benchmarks?
>
> Compared with actually running a standard benchmark, GPT-5+Search is far cheaper and faster, though less precise. Running a full benchmark—especially in pilot studies or during dataset curation—requires assembling the data, implementing the evaluation protocol, and performing potentially hundreds of model inferences, which can take days of effort and substantial GPU budget. In contrast, GPT-5+Search produces a rough performance estimate from **a single prompted prediction**, making it useful when the goal is rapid screening or prioritization rather than exact scoring.
>
> Thus, GPT-5+Search is not meant to replace ground-truth evaluation on established benchmarks; instead, it supplements the process by providing quick, low-cost estimates that help decide whether a proposed dataset or configuration is worth fully evaluating.

---

> ### Author Response · Authors · 2025-11-21
> **Response to Reviewer zA62 [2/2]**
>
> > Is there any heterogeneity predicting the performance across different benchmarks and different models?
>
> Yes, we analyzed heterogeneity on the PRECOG 2025 snapshot along several axes (model family/size, benchmark type, publication status, target model, and experimental vs benchmark setups).
>
> ---
>
> ***1. Benchmark characteristics (knowledge vs reasoning, math, coding, etc.)***
>
> **GPT-5 by benchmark category**
>
> | **Category** | **n** | **MAE ↓** | **Pearson r ↑** |
> | --- | --- | --- | --- |
> | Knowledge | 970 | 14.53 | 0.68 |
> | Reasoning | 1100 | 14.82 | 0.67 |
> | Math | 131 | 15.00 | 0.74 |
> | Coding | 170 | 15.11 | 0.73 |
> | Safety | 63 | 15.85 | 0.74 |
> | Instruction Following | 382 | 15.57 | 0.66 |
> | Multilingual | 60 | 12.07 | 0.78 |
>
> **Qwen3-32B by benchmark category**
>
> | **Category** | **n** | **MAE ↓** | **Pearson r ↑** |
> | --- | --- | --- | --- |
> | Knowledge | 971 | 19.35 | 0.41 |
> | Reasoning | 1101 | 19.59 | 0.37 |
> | Math | 131 | 21.43 | 0.45 |
> | Coding | 170 | 20.44 | 0.55 |
> | Safety | 63 | 19.62 | 0.53 |
> | Instruction Following | 382 | 20.74 | 0.40 |
> | Multilingual | 60 | 17.92 | 0.60 |
>
> For both predictors, the “Multilingual” category has the lowest MAE, which may reflect the relatively small variance of BLEU scores in machine translation benchmarks. In contrast, math and instruction-following tasks are harder to predict, likely because they allow more diverse task formulations and evaluation setups.
>
> ---
>
> ***2. Peer-reviewed vs non–peer-reviewed benchmarks***
>
> | **Model** | **Regime** | **n** | **MAE ↓** | **Pearson r ↑** |
> | --- | --- | --- | --- | --- |
> | GPT-5 | Peer-reviewed | 486 | 15.46 | 0.69 |
> | GPT-5 | Not peer-reviewed | 1036 | 14.50 | 0.68 |
> | Qwen3 | Peer-reviewed | 487 | 20.70 | 0.40 |
> | Qwen3 | Not peer-reviewed | 1036 | 19.10 | 0.42 |
>
> Performance is similar on peer-reviewed and non–peer-reviewed benchmarks for both models, and in fact is slightly better on non–peer-reviewed benchmarks, which is the opposite of what we initially expected.
>
> ---
>
> ***3. Target model (GPT-4 vs GPT-4o)***
>
> | **Predictor** | **Target model** | **n** | **MAE ↓** | **Pearson r ↑** |
> | --- | --- | --- | --- | --- |
> | GPT-5 | GPT-4 | 406 | 13.94 | 0.73 |
> | GPT-5 | GPT-4o | 1116 | 15.20 | 0.66 |
> | Qwen3 | GPT-4 | 407 | 18.13 | 0.48 |
> | Qwen3 | GPT-4o | 1116 | 20.25 | 0.40 |
>
> Both predictors perform better when forecasting performances of GPT-4 than GPT-4o, suggesting that newer or more variable targets are modestly harder to predict. One possible explanation is that many experimental results involving GPT-4 (compared to GPT-4o) were reported in 2023–2024 and are more likely to have influenced the training of GPT-5 and Qwen3.
>
> ---
>
> ***4. Experimental setups vs benchmark-introducing setups***
>
> | **Model** | **Regime** | **n** | **MAE ↓** | **Pearson r ↑** |
> | --- | --- | --- | --- | --- |
> | GPT-5 | Experimental setup | 973 | 14.72 | 0.69 |
> | GPT-5 | Benchmark setup | 549 | 15.12 | 0.67 |
> | Qwen3 | Experimental setup | 974 | 19.11 | 0.42 |
> | Qwen3 | Benchmark setup | 549 | 20.71 | 0.40 |
>
> Results are very similar between “experimental setup” entries (using existing benchmarks) and “benchmark setup” entries (introducing new benchmarks). This suggests that the predictor is not limited to canonical, long-established benchmarks, but can also handle newly introduced, previously unseen datasets. We have added this diverse-subset ablation study in Table 4.

---

### Official Review · Reviewer_7fFc · 2025-11-01

**Soundness:** 1
**Presentation:** 2
**Contribution:** 1
**Rating:** 2
**Confidence:** 4

**Summary:**

This paper investigates text-only performance forecasting: estimating a model’s score using only the task and configuration description. The authors collect a corpus and run experiments across multiple LLMs. Although I remain skeptical and the results largely support that skepticism, the authors argue that the task is feasible. They present this work as an initial step toward open-ended anticipatory evaluation, while I hope it is the final step.

**Strengths:**

- The paper studies a bold question.

**Weaknesses:**

- The task is unrealistic to me. An experiment's score depends on the test data, the model, and the testing configuration. I don't see how one could compress billions of parameters and gigabytes of data into a few sentences and still predict the resulting scores. The paper's results do not support the authors' claim, given a mean absolute error above 14.0%.
-  I see no practical use for so-called open-ended anticipatory evaluation. Most new methods and models aim for small, incremental improvements. An MAE of 14.0% conveys little meaningful information about performance gains or losses. If the forecasted score is 50%, the real score could lie anywhere between 36% and 64%.

**Questions:**

- Did you try your methodology on your own experiments? What's the estimated performance of all experiments in this paper?

---

> ### Author Response · Authors · 2025-11-21
> **Response to Reviewer 7fFc [1/2]**
>
> Thank you for your thoughtful feedback. We address each of your points below.
>
> > The task is too challenging.
>
> Our goal is to show that high-level experimental descriptions contain enough signal to yield useful, non-trivial performance estimates that can guide pilot design and dataset curation. We address this with three pieces of new evidence.
>
> ---
>
> **Leakage-free PRECOG 2025 results: description-based prediction is competitive and strongly correlated**
>
> We recognized after submission that our original setup had a leakage-related limitation, and we have been working since the deadline to address it. Accordingly, we have now re-run all experiments on a leakage-controlled PRECOG 2025 snapshot, using models whose knowledge cutoff is 2024 for both GPT-5 and Qwen3, ensuring that the snapshot postdates the models’ training data.
>
> | **Method** | **MAE ↓** | **Pearson r ↑** |
> | --- | --- | --- |
> | Estimation-set mean baseline | 23.0 | – |
> | E5-Mistral + kNN | 21.2 | 0.22 |
> | E5-Mistral + XGBoost | 21.4 | 0.20 |
> | GPT-5, medium reasoning | 14.6 | 0.52 |
> | GPT-5, minimal reasoning | 15.5 | 0.62 |
> | Qwen3-32B, with reasoning | 19.6 | 0.41 |
> | Qwen3-32B, no reasoning | 22.5 | 0.37 |
>
> - **Non-trivial predictive signal**: Even after further controlling for potential leakage, GPT-5 and Qwen3 *substantially* outperform both the estimation-set mean and embedding-based baselines in MAE and correlation. This shows that a description plus literature context still carries enough information to predict scores in a useful way.
> - **Role of reasoning**: For both GPT-5 and Qwen3, reasoning-style prompting improves robustness and correlation compared to weaker prompting (medium vs minimal reasoning, reasoning vs no reasoning), suggesting that the model is not simply memorizing but actively using the textual cues.
>
> We do not claim that a 14–15% MAE is perfect. Rather, we argue that this level of error, especially given that the MAE is substantially lower on the high-confidence subset (see Fig. 4) is sufficient for screening and prioritization in the pilot phase, which is the intended use illustrated below. We have added the new 2025 snapshot results to Tables 2 and 3.
>
> ---
>
> **Experimental vs benchmark entries and unseen datasets: task feasibility beyond “seen” settings**
>
> We also analyzed two regimes using PRECOG 2025. **Experimental *entries*:** instances where authors reuse benchmarks introduced elsewhere. **Benchmark *entries*:** instances where the paper *introduces* the dataset.
>
> | **Model** | **Regime** | **n** | **MAE ↓** | **Pearson r ↑** |
> | --- | --- | --- | --- | --- |
> | GPT-5 | Experimental records | 973 | 14.7 | 0.69 |
> | GPT-5 | Benchmark records | 549 | 15.1 | 0.67 |
> | Qwen3 | Experimental records | 974 | 19.1 | 0.42 |
> | Qwen3 | Benchmark records | 549 | 20.7 | 0.40 |
>
> These results show:
>
> - Comparable error and correlation across regimes. Performance is not confined to “easy” experimental setups; it generalizes similarly to benchmark-introducing papers.
> - For newly curated, truly unseen datasets from the 2025 arXiv snapshot, GPT-5 and Qwen3 achieve a Pearson correlation of 0.66, despite no exposure to those datasets during curation. This supports the feasibility of using description-based prediction for *new* datasets, not just ones already well-known.
>
> Taken together, these results suggest that while predictions are noisy, they are meaningfully aligned with true scores across different regimes and new datasets, which is exactly what we need to guide early-stage decisions. We have added these results to Table 4.d.
>
> ---
>
> **Contemporary related work: the community is converging on description-based prediction**
>
> We have added a citation to contemporaneous work recently accepted at NeurIPS [1], which similarly compresses research methods and experimental setups into short descriptions to predict outcomes (in their case, a binary score-based label). Their findings further show that training on such descriptions improves predictive accuracy.
>
> Our setting is similar but focused on dataset and experimental-setup–level prediction. This line of work collectively indicates that, while we cannot encode “billions of parameters and gigabytes of data” exactly, detailed, structured descriptions contain enough information to support practically useful performance prediction tasks. We have added this related work at Line 141.
>
> [1] Predicting Empirical AI Research Outcomes with Language Models, Wen et al, NeurIPS 2025
>
> ---
>
> In summary, we are *not* trying to replace full benchmark runs or detect 1–2 point improvements. The intended use is *early-stage triage*: distinguish obviously weak regimes (e.g., 30–40) from potentially deployment-ready ones (70–80), decide whether to invest in large-scale annotation, or choose which configurations to pilot.

---

> ### Author Response · Authors · 2025-11-21
> **Response to Reviewer 7fFc [2/2]**
>
> > Practical utility of open-ended anticipatory evaluation
>
> Our aim is to give a rough but useful signal about where to invest time, annotation budget, and compute before running costly sweeps. In that setting, an MAE in the mid-teens with non-trivial correlation is still valuable: it separates clearly bad regimes from plausibly viable ones, rather than trying to finely rank near-tied models.
>
> We would like to present a case study to illustrate the potential utility of our method. The puzzle-show benchmark [2] provides a concrete, quantitative case study for our approach in a setting where the original authors already suspected that off-air weekly challenges would be more difficult than on-air live puzzles. In this benchmark, GPT-4o achieves about 65% accuracy on the easier on-air subset and about 6% on the harder off-air subset. Given only natural-language task descriptions (e.g., "live, quickly solvable on-air puzzles answered in seconds by callers" vs. "harder, week-long off-air challenges that typical listeners may think about for days"), our predictor outputs 68% (on-air) and 14% (off-air) for GPT-4o, with absolute errors of 3 and 8 percentage points, respectively. The model correctly predicts a substantial performance drop on the off-air version before the actual off-air score was measured. In a hypothetical design scenario where neither subset has yet been constructed, a benchmark designer could write detailed descriptions of an "easy/live" variant and a "hard/take-home" variant, query our system once, and use the predicted gap (68 vs. 14) as ex-ante evidence that the harder off-air set is both tractable and non-saturated—helping them prioritize data collection and curation for that variant before running costly pilot experiments on both. We have included this limitation and scope in Section 9 (L511), and we provide a more detailed case study discussion in Appendix D.
>
> [2] PhD Knowledge Not Required: A Verbal Reasoning Challenge for Large Language Models, Wu et al., arXiv 2025
>
> | Subset | Gold accuracy (%) | Predicted accuracy (%) | Absolute error |
> | --- | --- | --- | --- |
> | On-air | 65 | 68 | 3 |
> | Off-air | 6 | 14 | 8 |
> | Overall | – | – | 5.5 (mean absolute error) |
>
> > Did you try your methodology on your own experiments?
>
> Yes. In Section 7 L 497, we included a self-referential evaluation where the system predicts its own performance on the task in this paper. GPT-5+Search predicted an MAE of 16.0 for itself, while the actual MAE reported in Table 3 is 14.0.

---

### Author Response · Authors · 2025-11-21
**General Response to Reviewers**

We thank all the reviewers for taking their valuable time to provide insightful comments and constructive suggestions to improve our paper.

We are pleased to note that the reviewers recognized the value of our work, as evidenced by the following positive comments:

- “the systems seem quite useful for accelerating the development and research of LLMs” (**zA62**)
- “clear differentiation from prior work” (**zA62**)
- “the paper studies a bold question” (**7fFc**)
- “presents an interesting framing of model evaluation as a forecasting problem” (**3Gxd**)
- “the paper considers an interesting setting, and the results look promising” (**EXyJ**)
- “the dataset appears well documented and carefully constructed, with redaction, leakage control, and validation steps” (**3Gxd**)
- “develop their own retrieval-based tool …, which both enriches the setup and helps ensure a fairer comparison across models” (**3Gxd**)
- “impressed by the streaming prediction and human baseline” (**zA62**)

We believe the thoughtful reviews and the constructive recommendations made by the reviewers have substantially improved the quality of the paper. Based on the reviewers' suggestions, we have updated the paper with the following major changes:

- Revised the Related Work section to be more concise added contemporary work published in NeurIPS 2025 (L140). (**zA62, 7fFc)**
- Reorganized Section 4, 5, 6 based on the PRECOG experiments using the 2025 arXiv snapshot to avoid further risk of data contamination. (**3Gxd, EXyJ, 7fFc)**
- Included a detailed experimental setup descriptions (e.g., decoding, k-fold cross validation) in Section 5 (L278, L285). (**EXyJ)**
- Added reasoning and non-reasoning mode ablation results into Table 3. (**3Gxd, EXyJ)**
- Incorporated an ablation study using a different subset of PRECOG in Table 4. (**zA62, 3Gxd, EXyJ)**
- Revised the text describing the human analysis to add limitation and discussion in Section 5 (L411). (**3Gxd)**
- Clarified and extended the discussion of why the retrieval module offers limited benefits for the main PRECOG experiments in Section 7 (L454). (**3Gxd)**
- Added GPT5 (without search) on Streaming Prediction Results in Table 7. (**EXyJ)**
- Added an illustrative example of a practical use case in Appendix D and a discussion of the benefits of the task in Section 9 (L511). (**7fFc, zA62, EXyJ, 3Gxd)**
- Included a results and discussion of cost efficiency of our approach in Section 9 (L521). (**zA62)**

We believe that these changes address your comments and improve the clarity and the quality of the paper. Thank you again for your thoughtful feedback and continued consideration of our work!

---

### Meta-Review · Area_Chair_fb8Z · 2026-01-05

**Summary:**

The authors frame model evaluation as forecasting: predict benchmark scores from redacted, schema-aligned descriptions without running the experiment, and releases PRECOG, a corpus of description-performance pairs mined from arXiv. Reviewers agreed the framing is interesting and the dataset curation/leakage controls are thoughtful, but raised major concerns about (i) practical utility given mid-teens MAE, (ii) leakage/contamination and whether gains reflect memorized benchmark patterns, (iii) baseline strength and reproducibility, and (iv) the role of retrieval and reasoning being unclear. In rebuttal, the authors strengthened the work by rebuilding a leakage-controlled PRECOG 2025 snapshot (post-cutoff), adding reasoning vs non-reasoning ablations, stronger embedding baselines, a streaming ablation without search, dataset-name recovery checks, and clearer positioning and usage examples. Ttwo reviewers indicated their concerns were mostly addressed and tentatively raised scores; one remained skeptical mainly on demonstrated utility. I think that this paper is on a great trajectory but might work better for a new submission to another conference.

**Reviewer Concerns:**

Addresse:

Leakage/contamination risk
Reproducibility / missing details
Baselines too weak
Practical positioning and example
Retrieval role unclear

Perhaps still missing:
Practical benefit not yet demonstrated
MAE may be too coarse for realistic ML decisions
Interpretability of what is being predicted is still somewhat ambiguous

**Reviewer Scores:**

7fFc: likely fro 2 to 2 because core skepticism about utility likely unchanged.
3Gxd: likely 2 to 3 acknowledged many issues addressed but still concerned about concrete utility + baseline strength.
EXyJ: very likely from 4 to 5 because they explicitly said most concerns addressed and tentatively raised score.
zA62: likely remained as a 6 because concerns mostly addressed but unlikely to go beyond marginal accept.

These are all guesses, of course, but it would lead to an average reject score.

---

### Decision · Program_Chairs · 2026-01-26

Reject